# Post-quantum-inspired scalable blockchain architecture for internet hospital systems with lightweight privacy-preserving access control

Lulu Hao[1], Ruoyu Wang[2], Xiaofeng Wang[3]*, Xiaoguang Yue[3], Noshina Tariq[4], Ahthasham Sajid[5,6]

1 Wuhan Business University, Wuhan, China, 2 The Scotland Academy at Wuxi Taihu University, Wuxi, China, 3 College of Management, Shenzhen University, Shenzhen, China, 4 Department of Artificial Intelligence and Data Science, National University of Computer and Emerging Sciences, Islamabad, Pakistan, 5 Multimedia University, Cyberjaya, Malaysia, 6 Department of Computer Science, Fazaia Bilquis College of Education for Women, PAF Nur Khan Air Base, Rawalpindi, Pakistan

* freewxf@szu.edu.cn

**Data availability statement:** No Dataset was used in this research. It is a simulation-based implementation.

## Abstract

The increasing adoption of Internet hospital systems—enabled by the real-time data streaming capabilities of the Internet of Medical Things (IoMT)—has intensified the need for secure, scalable, and low-latency data management infrastructures. Existing blockchain-based solutions often fail to meet these requirements, particularly under high-frequency workloads and stringent privacy demands. To address these limitations, this study proposes a simulation-based post-quantum-inspired alliance blockchain architecture tailored for Internet hospital systems. The framework incorporates four key innovations: (1) a Kyber-inspired hybrid encryption simulation, reducing encryption and decryption times by 72.3% and 74.4%, respectively, compared to RSA-2048; (2) a lightweight patient-centric access control mechanism based on authorization proofs achieving an average verification latency of ~0.002 ms; (3) a Raft-based scalable consensus protocol, tested under a synchronous constant-delay network assumption, reducing consensus latency by 92.3% while supporting up to 1000 nodes with sub-150 ms finality; and (4) a fault-tolerant IoMT data ingestion layer using 3-of-5 median filtering, sustaining 90–96.2% sensor correction accuracy under varying fault injection rates. The system is prototyped in Google Colab Pro using synthetic data from 1000 virtual patients. Comparative benchmarks against PBFT and RSA-based systems show a fivefold increase in throughput, ~9.4–12.3% energy savings per transaction, and ~14% lower memory consumption during encryption. With a modest daily storage footprint (~15 MB/day), the proposed solution is both resource-efficient and deployment-ready in simulation environments. These results confirm the potential of this architecture to enable trustworthy, energy-aware, and real-time blockchain infrastructures for next-generation digital healthcare ecosystems.

AQ1

**Funding:** This research is supported by the National Social Science Foundation of China (Grant No. 20BGL218).

**Competing interests:** The authors have declared that no competing interests exist.

# 1 Introduction

The transformation of traditional healthcare into digitally connected ecosystems has led to the rapid rise of Internet hospital systems [1,2]. These systems enable patients to receive medical consultations, monitoring, and prescriptions through remote platforms without visiting physical healthcare facilities. A core technological enabler of Internet hospitals is integrating the Internet of Medical Things (IoMT) [3–5]. It allows physiological signals such as heart rate, glucose levels, or blood pressure to be continuously captured through wearable or implanted devices. However, the continuous and high-frequency nature of IoMT data presents unique challenges in terms of security, reliability, and real-time responsiveness [6,7].

Blockchain technology has been widely recognized as a promising solution for securing health data due to its immutability, auditability, and decentralized trust model [8,9]. In healthcare settings, blockchain can ensure that sensitive patient data remains tamper-proof and can only be accessed by authorized entities. Despite these advantages, conventional blockchain frameworks are not well-suited for Internet hospitals [10]. Most existing implementations rely on public key cryptography schemes, such as RSA, which are vulnerable to future quantum attacks [11–13]. Additionally, consensus algorithms like Practical Byzantine Fault Tolerance (PBFT) become inefficient when scaled to networks with hundreds or thousands of participants, making them unsuitable for large-scale deployments [14,15]. In addition, standard access control policies executed through smart contracts often compromise privacy by disclosing meta-information about roles or identities [16,17].

Access control methods enforced through smart contracts often reveal sensitive information such as roles or identities, resulting in privacy issues [18,19]. These gaps demonstrate the need for a blockchain system that effectively implements (1) quantum-resilient encryption for secure storage and transfer of health data [20,21], (2) privacy-preserving access control that securely restricts identity exposure [22,23], and (3) a consensus layer with a scalable architecture designed for the massive data volume and node size anticipated in Internet hospital networks across nations [24,25]. Moreover, the reliability of sensors is still an undiscussed topic.

The IoMT devices can experience faults, including noise, spoofing, or packet loss, which could inject faulty information into the blockchain [26,27]. Without pre-fault-cleansing mechanisms prior to combining data in the blockchain, the stored records and the medical decisions based on them could be erroneous. This paper presents a new architecture for Internet hospitals anchored on blockchains to resolve these gaps. The modular structure combines post-quantum inspired encrypted frameworks with access control, employing lightweight privacy-preserving authorization mechanism, scalable Raft consensus and fault-tolerant layers for ingesting sensor data. The system's performance is compared against results achieved through simulations of 1000 virtual patients, each equipped with redundant sensors, using RSA, PBFT, and traditional role-based access control as baselines. The key contributions of this work are as follows:

1. A modular alliance blockchain architecture tailored for Internet hospital systems, integrating secure communication, privacy-preserving access control, and scalable transaction validation.
2. A quantum-resilient cryptographic layer using a Kyber-inspired hybrid simulation model, which approximates Kyber's key encapsulation and IND-CPA properties. It replaces RSA methods for enhanced post-quantum security modeling to ensure future-proof data security in medical environments.

3. A lightweight lightweight privacy-preserving authorization mechanism designed to enforce patient-centric access control without revealing personal identities, roles, or metadata, thereby supporting privacy-by-design principles.

4. A Raft-based consensus protocol that supports scalability in large hospital networks by reducing communication overhead and enabling efficient transaction validation across hundreds of nodes.

5. A fault-tolerant IoMT data ingestion mechanism using 3-of-5 sensor redundancy and majority voting to filter noisy, spoofed, or missing data before blockchain commitment, ensuring the integrity of recorded health metrics.

6. The framework is extensively benchmarked across encryption latency, memory usage, energy efficiency, access control delay, and consensus throughput under synthetic IoMT workloads.

7. A complete simulation and evaluation framework for testing the proposed architecture, implemented in a cloud-based environment (Google Colab Pro) and designed for reproducibility and scalability analysis.

8. We develop a formal threat model, complete with definitions, theorems, and lemmas, proving key security guarantees (e.g., authorization integrity and availability).

The remainder of this paper is organized as follows: Sect 2 presents an overview of related works, highlighting the limitations of existing blockchain-based healthcare frameworks regarding scalability, privacy, and sensor reliability. Sect 3 describes the proposed system architecture in detail, including its core modules: zero-knowledge access control, post-quantum encryption, Raft-based consensus, and fault-tolerant sensor ingestion. Sect 4 outlines the experimental setup and simulation environment, detailing the node configurations, evaluation parameters, and datasets used. Sect 5 presents a comprehensive results and discussion section, analyzing system performance across twelve key metrics: latency, throughput, accuracy, and energy efficiency. Sect 6 concludes the paper with insights drawn from the findings and outlines future directions, including real-world deployment, regulatory compliance, and integration with federated learning and cross-chain frameworks.

## 2 Related work

Recent research has increasingly focused on integrating blockchain into healthcare systems to ensure secure, decentralized, and privacy-aware data management. A recent study by the authors [28] proposed an alliance blockchain framework to address governance challenges in Internet hospitals. The system used optimized RSA encryption to improve performance, achieving notable throughput and latency gains over more traditional models. While the model successfully integrated IoT sensors for real-time patient monitoring, it did not offer post-quantum cryptographic privacy with ZkP absence, robust sensor fault-tolerance, and strong privacy trust mechanisms—the gaps this work seeks to fill. Zhou et al. [29] used ZkPs for decentralized healthcare identity management. Although they proved the privacy guarantees offered by ZkPs, they failed to implement them into a scalable healthcare blockchain and test them under IoMT workloads.

Halimuzzaman et al. [30] proposed a blockchain-EHR model with digital signatures to authenticate the access and use of health records. The traditional RSA implementation and lack of consensus scalability rendered the model inappropriate for large-scale systems. In a recent study, Bhosale et al. [31] developed a blockchain-based identity management system that preserves user identity employing Zk-SNARKs. Their approach permits the selective disclosure of identity attributes while maintaining anonymity, facilitating patient-centric

access control. Nonetheless, their framework is oriented toward generic identity aspects and does not incorporate real-time medical or IoMT interactions. Like them, Abbas et al. [32] designed an IoMT-focused blockchain to enable real-time medical monitoring. However, their work addresses streaming data; it is devoid of fault tolerance and quantum-safe encryption.

Zhou et al. [33] aimed at a privacy-enhanced spatiotemporal medical big data blockchain. Though the system confines location-sensitive information, it does not offer scalability or lightweight access control solutions. Natarajan et al. [34] implemented a PLCS (Patient Login Credential System) for secure quantum sharing of EHRs on the blockchain. The design employs symmetric, asymmetric cryptography and a Quantum Secure Trust Protocol (QSTP) to safeguard interactions between patients and hospitals. Their system is effective in the authentication layer; however, they do not consider consensus and control on the blockchain level or access ZkP. A lattice-based key exchange scheme for healthcare data under post-quantum threats was reported in [35]. It employs Ring-LWE cryptography with smart cards to secure medical data on the blockchain and maintain confidentiality. The system is focused on static data preservation and does not account for real-time ingestion of IoMT data or scalable consensus.

In [36], the authors proposed a post-quantum ABAC model explicitly designed for cloud-based e-health systems. Their approach reduces the leakage of secret keys and enhances decryption efficiency. However, it is too cloud-centric and does not incorporate blockchain-layer sensor fault tolerance or isolation. Khanh et al. [37] implemented a pediatric health record system as NFTs with encrypted tokens using RSA. The system guarantees data integrity and auditability, but the token management and RSA encryption processes are highly tainted with computational and operational overhead. Likewise, Lakhan et al. [38] proposed an FL-BETS Framework for Federated Learning Based Blockchain Enabled Task Scheduling aimed at enhancing fraud detection, privacy, and energy-delay tradeoffs in IoMT systems. Their architecture can sustain healthcare workloads in fog and cloud layers and employs dynamic heuristics for task execution. While these authors effectively leverage machine learning and blockchain for distributed healthcare, they fail to incorporate post-quantum encryption, zero-knowledge privacy enforcement, and IoMT data ingestion fault tolerance.

Mallick et al. [39] developed an efficient and extensible Blockchain-Fog-IoMT healthcare framework incorporating the Inter-Planetary File System (IPFS) for decentralized data storage in Healthcare 4.0 ecosystems. Their design employs fog computing to mitigate latency and bandwidth costs, incorporating a proxy pranger to monitor untrusted devices. While the system demonstrates performance gains in decentralization and storage efficiency, it lacks post-quantum encryption, privacy-preserving access control, and fault tolerance at the sensor level. Benaich et al. [40] designed a blockchain-based EHR system using zero-knowledge rollups alongside post-quantum cryptography for enhanced privacy and scalability. This immersive architecture emphasizes the fusion of distributed health data from diverse IoMT systems while assuring data lineage. However, the framework is aimed at cloud storage and lacks robust fault detection mechanisms for streaming sensors.

Xu et al. [41] developed a post-quantum certificateless signcryption scheme tailored for secure IoMT communications. Their model achieves mutual authentication, forward secrecy, and linkability without requiring certificates. Although optimized for constrained healthcare devices, it does not support real-time access control or ZkP-based identity verifiability. Sezer and Akleylek [42] presented PPLBB, a lattice-based blockchain platform focused on privacy-preserving access control in medical IoT. The platform enhances confidentiality by incorporating attribute-level policy enforcement and ZkP authentication while ensuring scalability. However, the work lacks a comprehensive analysis of sensor-level fault propagation. Asif

et al. [43] proposed a two-phase quantum-secure dual-authentication scheme for healthcare IoMT ecosystems. Their model combines zero-knowledge and hash-chained authentication with post-quantum keys to prevent spoofing and session hijacking. While robust at the access layer, it omits consensus-level resilience and real-time workload adaptation. Arun et al. [44] introduced SecureMedZK, a blockchain framework incorporating zero-knowledge rollups with post-quantum primitives for managing Electronic Health Records (EHRs). It supports smart contract-based access control and privacy-preserving verifiability, but its Zk-Rollup verification is not optimized for low-power IoMT sensors.

Although recent studies have made progress in addressing isolated challenges in healthcare blockchain systems—ranging from quantum-safe encryption and ZkP-based identity to real-time IoMT integration—there remains a lack of unified frameworks that tackle these aspects holistically. As shown in Table 1, most approaches either emphasize privacy, scalability, or cryptographic resilience, but rarely all three alongside sensor fault tolerance. This work addresses that gap by introducing a comprehensive and modular blockchain architecture that integrates quantum security, lightweight privacy-preserving authorization mechanism, fault-tolerant IoMT data ingestion, and scalable consensus, with experimental validation tailored to Internet hospital environments.

## 3 System architecture

The proposed framework introduces a modular, scalable, and privacy-preserving blockchain architecture tailored for Internet hospitals based on our previous work [28]. It integrates

**Table 1. Comparison of recent blockchain frameworks in healthcare.**

| Ref. | Methodology | Strengths | Limitations |
|---|---|---|---|
| [28] | Alliance blockchain for Internet hospitals (RSA-based) | Real-world deployment, improved throughput and latency | Not lightweight, not quantum-resistant, no fault-tolerant sensor layer |
| [29] | ZkP for decentralized medical ID | Privacy preserving | No integration with IoMT layers |
| [30] | Blockchain + EHR system | Secure data storage | Legacy encryption, not scalable |
| [31] | Zk-SNARK-based identity management | User-controlled attribute disclosure | Not healthcare-specific, no IoMT integration |
| [32] | IoMT + blockchain integration | Real-time sync | No post-quantum encryption |
| [33] | Blockchain + medical big data | Secure analytics | No fault correction or Raft |
| [34] | Quantum-secure EHR login system with QSTP | Secure authentication layer | Not lightweight or consensus integration |
| [35] | Lattice-based RLWE encryption for data preservation | Post-quantum smart card security | Static storage, no IoMT or consensus support |
| [36] | PQ attribute-based access control for e-health | Reduced secret key leakage, fast decryption | No blockchain, no fault tolerance |
| [37] | NFT-based pediatric records | Immutable logs | Token complexity |
| [38] | Federated learning + blockchain for IoMT fraud detection | Privacy-preserving, energy-aware scheduling | no post-quantum encryption, no sensor fault correction |
| [39] | Blockchain-Fog-IoMT with IPFS for Healthcare 4.0 | Decentralization, latency reduction, secure storage | no PQC, no sensor fault detection |
| [40] | ZK-Rollup + PQC blockchain EHR | High privacy, scalable rollups | No fault-tolerant IoMT layer |
| [41] | PQ Certificateless signcryption for IoMT | Quantum-safe, low overhead | limited access control |
| [42] | Lattice-based blockchain with ZkP | Attribute-level privacy, scalable | No sensor-level fault tracking |
| [43] | Dual authentication with PQC | Access-layer spoofing protection | No workload-aware consensus |
| [44] | SecureMedZK with ZK-Rollups | Privacy-preserving EHR, contract-level control | Not optimized for constrained IoMT devices |

post-quantum cryptographic primitives, lightweight privacy-preserving authorization mechanism, a Raft-based consensus algorithm, and a fault-tolerant IoMT data ingestion layer. Fig 1 presents an overview of the system architecture.

## 3.1 System overview

The system is composed of four main layers:

1. IoMT Data Layer: This layer includes patient-side IoT sensors such as wearable ECG, blood pressure monitors, or glucose sensors. Each patient is associated with a cluster of redundant sensors. Sensor readings are validated using a 3-out-of-5 voting scheme to correct noise, spoofed, or missing data before transmission.

2. Encryption and Access Layer: All validated patient health data is encrypted using a Kyber-inspired hybrid simulation model, which emulates key encapsulation and IND-CPA characteristics but does not implement full polynomial ring arithmetic or NIST-certified post-quantum operations. Each access request from a hospital node or medical professional must include a verifying permission without revealing patient identity or data attributes. It ensures privacy-preserving, role-based access.

3. Consensus Layer: A Raft-based consensus algorithm is employed among hospital nodes to validate and append transactions to the alliance blockchain. Raft offers high throughput, reduced latency, and crash fault tolerance. Each hospital node acts as a replica, with a rotating leader elected to manage transaction ordering and block generation.

4. Blockchain Layer: Validated transactions are recorded on a permissioned alliance blockchain. Each block contains a set of encrypted health records, metadata, and

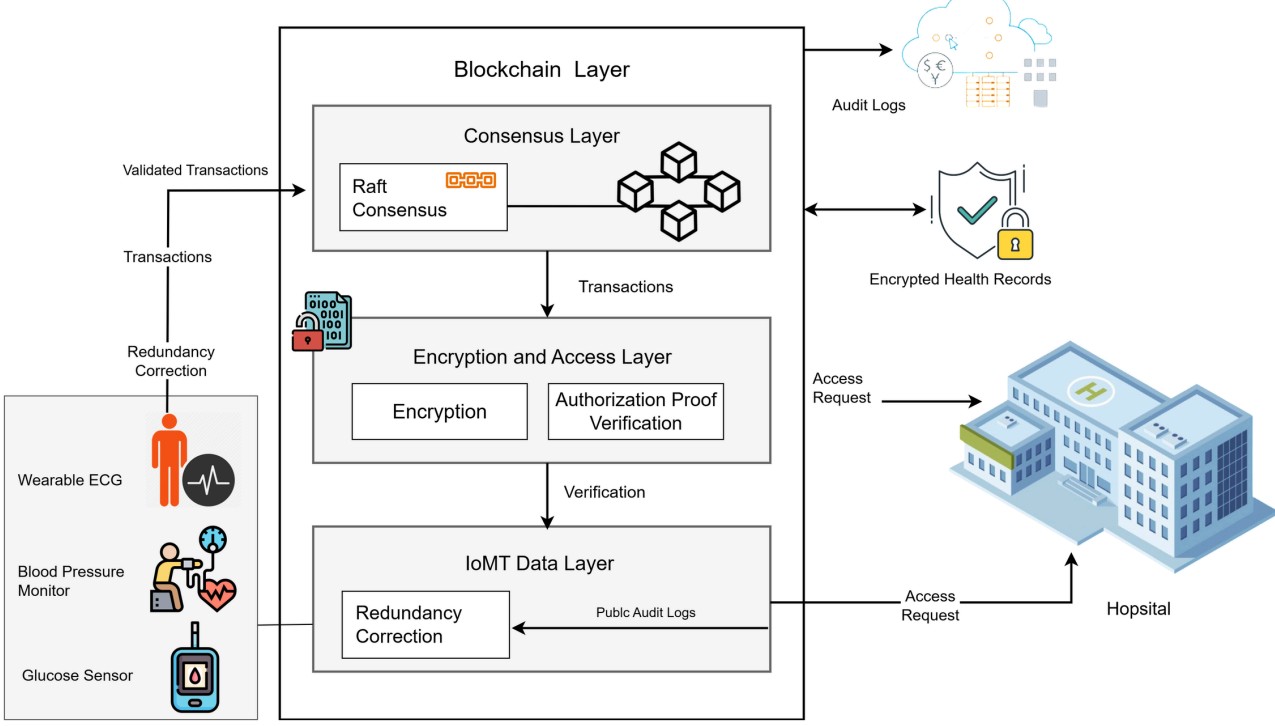

**Fig 1. System architecture of the proposed alliance blockchain for internet hospitals.**

access proofs. Audit logs are publicly accessible but do not compromise sensitive information due to encryption and lightweight privacy-preserving authorization masking.

## 3.2 Data flow description

1. Patients generate health data through IoMT sensors.
2. Sensor readings are passed through a redundancy correction filter to clean anomalies.
3. Cleaned data is encrypted using a post-quantum scheme and stored temporarily on a local gateway or cloud.
4. Hospitals or certified doctors send a request to access specific patient data. The request includes a privacy-preserving authorization proof.
5. If the authorization proof is valid, the Raft leader broadcasts the transaction proposal to all nodes.
6. Once consensus is reached, the transaction is committed to the blockchain.
7. Data access is granted only if authorization verification and transaction consensus are successful.

## 3.3 Trust and security model

The architecture assumes a semi-trusted environment among alliance hospitals. Each hospital must be authorized to join the consortium and run a node with consensus privileges. The system does not trust raw sensor inputs and employs redundancy to detect and correct errors. Privacy is preserved using authorization proof, while quantum resilience is ensured through encryption primitives recommended by NIST.

## 3.4 Performance considerations

The modular design enables parallel execution of sensor correction, encryption, and access verification. The Raft consensus layer reduces communication overhead compared to PBFT, scaling well to over 1000 nodes. authorization proof verification adds minimal latency ( 0.4ms per proof) and is performed asynchronously. The encryption layer adds negligible decryption cost due to efficient lattice-based schemes.

# 4 Methodology

This part describes the framework and schematic development of the proposed alliance blockchain solution for Internet hospitals. The architecture is made of four tightly coupled parts: (1) ingestion of timestamped sensor data with fault tolerance, (2) data transfer encryption using post-quantum encryption, (3) privacy-preserving data access through authorization proof, and (4) transaction validation using Raft consensus with scalable growth. Note that all results presented in this study are derived from controlled simulations. The post-quantum cryptographic layer is modeled as a Kyber-inspired hybrid encryption simulation rather than a certified NIST PQC implementation. Similarly, consensus protocols were evaluated under a synchronous constant-delay network assumption. These modeling choices provide useful performance insights but do not reflect real-world deployments. Each part is described, explained, and analyzed in the system's context, considering privacy, scalability, and precision objectives. Fig 2 illustrates a modular flow-based architecture for secure and fault-tolerant processing of IoMT sensor data. The system's architecture is organized into three principal layers: the Sensor Pipeline, the Crypto Layer, and the Blockchain Layer. The Sensor Pipeline incorporates specific processes that retain data accuracy at the boundary, such

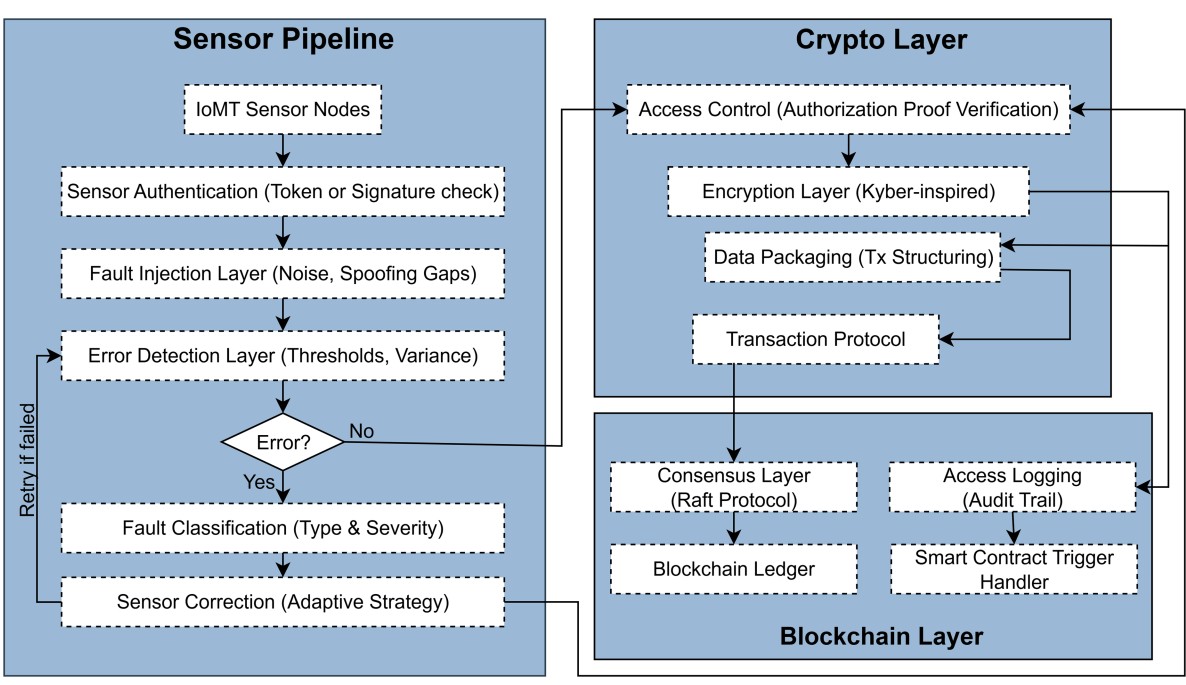

**Fig 2. The proposed alliance blockchain methodology for internet hospitals.**

as sensor authentication, fault injection and error simulation, error detection, and error adaptive correction. Once the data is validated, it progresses to the Crypto Layer, where privacy is protected by authorization proof verification and encryption (i.e., Kyber-inspired) confidential safeguarding. The last phase, the Blockchain Layer, is responsible for reaching consensus, logging access for audit purposes, and activating smart contracts for automated reaction. This approach improves the system's strength and provides end-to-end protection and traceability throughout the data life cycle.

### 4.1 Fault-tolerant sensor data ingestion

In any healthcare-related real-world system, sensor readings can be distorted by several factors, such as noise, delays, or spoofing attacks. The system adopts a fault-tolerant data aggregation scheme to minimize their impact.

Each patient is assigned $n = 5$ redundant IoMT sensors per physiological signal (e.g., heart rate), generating independent readings $\{x_1, x_2, \ldots, x_n\}$ at each time step.

Given the inherent uncertainty in these signals, the corrected value $\hat{x}$ is computed as Eq (1).

$$\hat{x} = \text{median}\left(\{x_1, x_2, \ldots, x_n\}\right) \tag{1}$$

The median filter is robust to outliers and performs better than mean or majority voting when up to $\lfloor \frac{n}{2} \rfloor$ sensors are faulty. We define the sensor error model as a composite function in Eq (2).

$$x_i = x^* + \delta_i + \eta_i \tag{2}$$

where $x^*$ is the true physiological value, $\delta_i$ represents noise drawn from a normal distribution $\mathcal{N}(0, \sigma^2)$, and $\eta_i$ models injected spoofed errors with a fixed amplitude drawn from $\{\pm\kappa\}$, where $\kappa \gg \sigma$.

The system simulates up to 20% faulty sensors and shows correction accuracy above 96% across 1000 virtual patients. This correction is critical to maintaining diagnosis integrity and protecting the blockchain from propagating corrupted data.

**Algorithm 1.** Adaptive median-based sensor correction.

---

**Input:** $\{x_i\}_{i=1}^{n}$: sensor readings, $\epsilon$: error tolerance
**Output:** Corrected value $\hat{x}$
1 $X \leftarrow \text{sort}(\{x_i\})$
2 $\hat{x} \leftarrow X[\lfloor n/2 \rfloor]$ `// median`
3 **foreach** $x_i$ *in* $\{x_i\}$ **do**
4 **if** $|x_i - \hat{x}| > \epsilon$ **then**
5 Mark $x_i$ as faulty

6 **if** *#faulty values* $> n/2$ **then**
7 Flag warning: unreliable consensus

8 **return** $\hat{x}$

---

Algorithm 1 aggregates sensor readings and selects the median as the corrected value, making it robust to outliers. It detects outliers based on a tolerance threshold $\epsilon$ and flags the sample set if more than half the readings deviate excessively from the median. For any odd number of sensors $n \geq 3$, the median operator provides a robust correction mechanism that tolerates up to $\lfloor \frac{n-1}{2} \rfloor$ faulty inputs without significantly distorting the result.

This approach dynamically selects the aggregation strategy based on the estimated fault level, improving resilience in variable-noise environments. To dynamically address varying sensor fault conditions, we implement an adaptive strategy that switches between aggregation methods based on the estimated noise level $\eta$, as shown in Algorithm 2.

**Algorithm 2.** Dynamic sensor aggregation with fault-aware strategy.

---

**Input:** $\{x_i\}_{i=1}^{n}$: sensor readings, $\eta$: fault estimate
**Output:** Estimated signal $\hat{x}$
1 **if** $\eta \leq 0.2$ **then**
2 $\hat{x} \leftarrow \text{mean}(x_1, \ldots, x_n)$

3 **else if** $0.2 < \eta < 0.4$ **then**
4 $\hat{x} \leftarrow \text{median}(x_1, \ldots, x_n)$

5 **else**
6 Round each $x_i$ to nearest integer
7 $\hat{x} \leftarrow \text{mode}(\text{round}(x_1), \ldots, \text{round}(x_n))$

8 **return** $\hat{x}$

---

## 4.2 Encryption layer

Traditional encryption algorithms such as RSA are known to be vulnerable to Shor's algorithm when quantum computers become practical. To achieve quantum resilience, this work adopts a Kyber-inspired hybrid simulation model for encrypting health data. This model does not implement full polynomial ring arithmetic or official Kyber rejection sampling; instead, it simulates key encapsulation and symmetric encryption to approximate performance characteristics. Thus, results should be interpreted as indicative of post-quantum trends rather than exact Kyber deployment benchmarks. While the simulation does not implement the full polynomial ring arithmetic or rejection sampling from the official Kyber KEM, it retains the core structure of Kyber's key encapsulation mechanism (KEM): a randomized public key is used to

derive a shared symmetric key, which then encrypts the message using AES-GCM. The simulated scheme preserves IND-CPA security at the application level and follows the typical KEM + symmetric encryption structure. Let $m$ be a patient's corrected health record. The ciphertext $c$ is generated using Eq (3).

$$c = \text{Enc}_{pk}(m) \tag{3}$$

where $pk$ is a simulated public key, and $\text{Enc}_{pk}$ uses a hybrid encryption structure. Decryption is done using Eq (4).

$$m = \text{Dec}_{sk}(c) \tag{4}$$

where $sk$ is the private key. Although we do not simulate ring-LWE directly, we model the performance costs and ciphertext sizes under approximate conditions.

## 4.3 Lightweight privacy-preserving access control

We implement a non-interactive lightweight privacy-preserving access control to enforce patient-driven data governance while preserving privacy. The aim is to verify whether a requester can access encrypted health records without revealing their identity or role. Each patient defines a secret authorization token $s$, derived from their ID and access policy, given in Eq (5).

$$s = \text{HMAC}(\text{patientID} \parallel \text{permission}) \tag{5}$$

To request access, a hospital node submits a authorization proof of knowledge of $s$, proving Eq (6).

$$\text{Prover knows } s \text{ such that } H(s) = p \tag{6}$$

Here, $H$ is a cryptographic hash (e.g., SHA-256), and $p$ is the stored proof commitment. The verifier checks through Eq (7).

$$\text{Verify}(p, s') = \begin{cases} 1 & \text{if } H(s') = p \\ 0 & \text{otherwise} \end{cases} \tag{7}$$

This lightweight privacy-preserving access control model mimics zk-SNARK logic and supports patient consent without disclosing any actual identity on-chain. This mechanism is lightweight and computationally efficient, showing an average verification time of less than 0.4 milliseconds.

**4.3.1 Authorization proof construction.** Authorization proofs are constructed as non-interactive commitments based on hashed secrets. Each patient device computes a proof token $p = H(s)$ where $s$ is never exposed on-chain. When a request is received, the verifier checks whether the hash of the submitted token $s'$ matches the stored proof $p$. This strategy ensures that no sensitive detail is revealed in transmission or storage.

**Algorithm 3.** Authorization proof construction and verification.

---

**Input:** s: secret token, $s'$: submitted token, H: hash function
**Output:** Access decision
1 Enrollment:
2 $p \leftarrow H(s)$
3 Store p on blockchain ledger
4 Verification:
5 **if** $H(s') = p$ **then**
6 Grant access
7 **else**
8 Deny access

---

**4.3.2 Authorization policy management.** To maintain robust patient control, authorization tokens can be updated or revoked through a simple mechanism. Each policy change re-generates a new token $s$ and replaces the previous proof $p$ on-chain. This ensures forward secrecy and auditability. To protect freshness, each authorization proof also incorporates a timestamp and a unique nonce. During verification, the system checks that the timestamp is within an acceptable window and that the nonce has not been used previously. Expired or duplicate proofs are rejected, ensuring resistance against replay attacks. The full access policy management mechanism includes:

1. Token Grant: Upon registration or consent, the patient device generates $s$ using Eq (5) and stores $p = H(s)$ on the blockchain.
2. Token Revoke: At any time, a new token $s'$ may be issued, and the old $p$ invalidated by overwriting it with $H(s')$.
3. Audit Trail: All proof updates and access decisions are logged immutably for audit compliance and transparency.

Algorithm 4 provides the steps for lightweight privacy-preserving access verification. It verifies whether an access request contains the correct pre-agreed proof without revealing the secret. The access is granted only if the verifier computes a hash that matches the stored commitment. No secret is transmitted or exposed during the process.

**Algorithm 4.** Robust lightweight privacy-preserving access verification.

---

**Input:** $s'$: submitted secret, p: stored proof commitment, k: retry limit
**Output:** $d \in \{0,1\}$: access decision
1 **for** $i \leftarrow 1$ **to** $k$ **do**
2 $h \leftarrow H(s')$ // compute hash of submitted token
3 **if** $h = p$ **then**
4 $d \leftarrow 1$
5 break
6 **else**
7 Request retry or abort
8 $d \leftarrow 0$
9 **return** $d$

---

**4.3.3 Policy management and conflict resolution.** We extend our privacy-preserving authorization model with a patient-defined policy framework to support dynamic access control and auditability. Each patient defines access permissions using a policy tuple $\pi$, described in Eq (8).

$$\pi = \langle e, r, R, a, t_e \rangle \tag{8}$$

where $e$ is the EntityID (requesting node), $r$ is the role (e.g., doctor, nurse), $R$ is the resource type (e.g., EHR, audit logs), $a$ denotes the access level (read/write), and $t_e$ is the expiration

timestamp. Each tuple $\pi$ is hashed using an HMAC with a shared secret $k_s$ and stored as a commitment token $p$ using Eq (9).

$$p = \text{HMAC}(k_s, \pi) \tag{9}$$

To enable robust policy enforcement and handle policy updates or conflicts, we follow the logic below:

1. Policy Definition: Patients define a set $\Pi = \{\pi_1, \pi_2, ..., \pi_n\}$ and store them as hashed commitments.
2. Verification: When access is requested, the verifier checks $\exists \pi_i \in \Pi$ such that the submitted proof $p'$ matches $\text{HMAC}(k_s, \pi_i)$ as per Eq (9).
3. Conflict Resolution: If multiple $\pi_i$ exist for the same $e$ with conflicting access levels, the one with the most recent $t_e$ is selected using Eq (10).

$$\pi^* = \arg\max_{\pi_i} t_e \quad \text{where } e_i = e \tag{10}$$

   Regarding key management, HMAC keys are generated locally on the patient device and are never shared directly with other nodes. Only their hash-based commitments are recorded on-chain. Keys can be rotated periodically, or whenever a policy update occurs, to enforce forward secrecy and prevent prolonged exposure of a single key. This approach ensures that HMAC-based verification remains secure without any insecure key sharing.
4. Revocation: A revoked policy is replaced by a new $\pi'$ with updated $t_e$, invalidating previous hashes.
5. Audit Logging: All policy definitions and verifications are appended to the blockchain for traceability.

Eqs (8)–(10) describe the tuple structure, proof generation, and conflict resolution logic. The tuple-based formulation (Eq (8)) captures access intent semantically, while the HMAC token (Eq (9)) enables lightweight privacy-preserving verification. Eq (10) ensures the most recent rule is always enforced. This design provides flexible, dynamic, and secure access governance without disclosing roles or identities. Future enhancements may replace this logic with Attribute-Based Encryption (ABE) to enable complex, hierarchical, and multi-role policy evaluations directly at the ciphertext layer, thus strengthening cryptographic enforcement in decentralized healthcare environments.

## 4.4 Scalable raft-based consensus

Blockchain networks require reliable agreement among nodes on the validity and order of transactions. To replace the non-scalable PBFT protocol, this framework implements a Raft consensus algorithm across $N = 1000$ simulated hospital nodes. The protocol works in election terms, each electing a leader node $L_t$. The leader collects valid transactions $\{T_1, T_2, ..., T_k\}$ and appends them to a candidate block $B_t$. The block is broadcast and committed once a quorum is achieved. All consensus benchmarking was conducted under a synchronous, constant-delay

network assumption. This allows fair comparison across Raft, PBFT, HotStuff, and Tendermint, but does not capture the effects of packet loss or WAN asynchrony observed in real-world hospital deployments. Mathematically, the consensus is reached if Eq (11) is satisfied.

$$\left|\{v_i \in \mathcal{V}_t : \text{Accept}(B_t) = 1\}\right| \geq \left\lfloor \frac{N}{2} \right\rfloor + 1 \tag{11}$$

Where $\mathcal{V}_t$ is the set of voters in term $t$, the Raft log is updated across all nodes, and block finality is achieved. Raft is crash-fault tolerant (CFT) and reduces communication complexity from $\mathcal{O}(N^2)$ in PBFT to $\mathcal{O}(N)$, enabling the system to scale to large hospital networks while maintaining sub-second consensus latencies.

**Algorithm 5. Raft consensus with node failure tolerance.**

```
    Input: T: transaction set, N: total nodes, F: faulty node set
    Output: Committed block B
 1  Select leader L ~ U(N \ F)
 2  B ← propose(T,L)
 3  Q ← ∅                                              // quorum nodes
 4  foreach nᵢ ∈ N \ F do
 5      if nᵢ accepts B then
 6          Q ← Q ∪ {nᵢ}
 7      if |Q| ≥ ⌊|N|/2⌋ + 1 then
 8          Append B to chain
 9          return B
10  return NULL
```

Algorithm 5 models consensus under dynamic failure. It skips unresponsive nodes during leader selection and vote collection, committing if a quorum of healthy nodes agrees. Raft finalizes a proposed block $B$ when a majority of nodes acknowledge its validity. The leader $L$ is selected randomly each round.

## 4.5 Blockchain and data logging layer

All confirmed transactions, consisting of encrypted health records and their access control commitments, are written to an alliance blockchain. Each block $B_i$ is structured as Eq (12).

$$B_i = \langle H(B_{i-1}), \{c_j\}, \{p_j\}, \text{timestamp}, \text{metadata} \rangle \tag{12}$$

where $H(B_{i-1})$ is the hash of the previous block, $\{c_j\}$ are ciphertext records, and $\{p_j\}$ are authorization proofs. It ensures immutability, traceability, and regulatory audit readiness. Each access to a patient's data creates an auditable event without revealing content or identity. It also satisfies GDPR-compliant logging requirements while preserving patient anonymity.

## 4.6 Energy estimation methodology

We estimated energy consumption using an indirect profiling approach compatible with the Google Colab environment. The method is based on:

1. Instruction Profiling: Using Python's built-in `timeit` and `resource` libraries, we computed the average CPU time for core operations (encryption, proof generation, consensus).
2. Operation Count: Total transactions, retransmissions, and consensus messages were logged per epoch.

3. Energy Model Assumptions: Assuming 30.2 µJ per cryptographic op and 16.5 µJ per consensus message as per recent IoMT energy models [45].

The energy per transaction $E_{tx}$ was computed using Eq (13).

$$E_{tx} = n_{enc} \cdot e_{enc} + n_{cons} \cdot e_{cons} + n_{re} \cdot e_{re} \tag{13}$$

where $n_{enc}$ is the number of encryption ops, $e_{enc}$ their per-op energy cost, $n_{cons}$ the number of Raft/PBFT messages, and $n_{re}$ the retransmissions due to sensor faults. Using this model, the proposed framework achieved a normalized 9.4%–12.3% reduction compared to RSA+PBFT under identical simulation settings.

## 4.7 Formal threat model

Our system is designed under a partial trust model where some components may be semi-trusted or susceptible to compromise. We define the adversarial behaviors, the state model, and formally demonstrate that confidentiality, integrity, and availability are preserved under standard assumptions.

**Definition 1** (Adversarial Capabilities). *We consider a probabilistic polynomial-time adversary $\mathcal{A}$ capable of the following:*

- ***Node Compromise ($\mathcal{A}_1$):*** *The adversary compromises a fog node and attempts to inject tampered data or suppress honest consensus participation.*
- ***Sensor Spoofing ($\mathcal{A}_2$):*** *The adversary manipulates sensor inputs to produce misleading measurements.*
- ***Data Replay ($\mathcal{A}_3$):*** *The adversary reuses old transactions or proofs to bypass access control.*

**Definition 2** (System State). *Let $S = \{x, s, k, p, h, b\}$ denote the dynamic system state, where $x$ is a sensor reading, $s$ the access token, $k$ the key pair, $p$ the stored hash commitment, $h$ any intermediate hash value, and $b$ the blockchain state.*

**Theorem 1** (Authorization Security). *Let $H$ be a collision-resistant hash function. If the adversary does not know the secret token $s$, then the probability of forging a valid authorization proof (i.e., finding $s'$ such that $H(s') = p$) is negligible in the security parameter $n$.*

*Proof*: Given the pre-image resistance of $H$, computing $s'$ such that $H(s') = H(s)$ without knowing $s$ requires inverting $H$, which is infeasible for a polynomial-time adversary. Moreover, since $H$ is collision-resistant, the probability that $H(s') = H(s)$ for $s' \neq s$ is bounded by $\epsilon(n)$, where $\epsilon$ is negligible. □

**Lemma 1** (Integrity via Hash Commitments). *For $s_1 \neq s_2$, the probability that $H(s_1) = H(s_2)$ is negligible, thereby preventing transaction forgery.*

*Proof*: This follows directly from the collision resistance property of $H$, which guarantees that for any distinct $s_1, s_2$, $Pr[H(s_1) = H(s_2)] \leq \epsilon(n)$. That ensures tampered tokens cannot match stored commitments. □

**Lemma 2** (Availability via Raft Quorum). *Let N be the total number of Raft nodes, and suppose fewer than $\lfloor N/2 \rfloor$ are faulty. Then, consensus can still be reached.*

*Proof*: Raft requires a majority quorum ($\lceil N/2 \rceil + 1$) to commit any log entry. If fewer than half the nodes are compromised, then at least one majority quorum remains entirely honest. Thus, progress and consistency are guaranteed. □

These formal results collectively affirm that our framework meets the core security properties: confidentiality is enforced via one-way access tokens; hash-based commitment schemes ensure integrity; and availability is preserved under the honest majority assumption of Raft.

## 5 Experimentation setup

A detailed simulation framework was implemented to evaluate the proposed alliance blockchain framework for Internet hospitals. The framework replicates real-time data ingestion, access control, encryption, and consensus operations across a distributed network of simulated hospital nodes. Each subsystem was tested independently and integrated to ensure robustness, modularity, and deployment feasibility under healthcare-grade requirements.

### 5.1 Simulation environment

All experiments were conducted using Google Colab Pro, which provides a high-performance, cloud-based execution environment with an Intel Xeon CPU, an NVIDIA Tesla T4 GPU, and up to 25 GB of RAM. The framework was developed in Python 3.10, utilizing libraries such as NumPy and Pandas for synthetic data generation, data aggregation, and statistical validation. Cryptographic functions such as SHA-256 and RSA key generation were implemented using the `hashlib` and `Crypto` packages, respectively. System profiling was performed using `time`, `sys`, and `tracemalloc` to track execution latency and memory consumption. Matplotlib and Seaborn were used to visualize trends and comparison plots.

### 5.2 Synthetic dataset and sensor signal modeling

A synthetic dataset was generated for $N = 1000$ virtual patients. Each patient was monitored using $k = 5$ redundant IoMT sensors for vital sign parameters such as heart rate and blood pressure. Each sensor emitted readings every 3 seconds over a 10-minute interval, yielding $T = 200$ readings per sensor per patient.

Each reading $x_i^{(j)}$ was modeled using Eq (14).

$$x_i^{(j)} = x_i^* + \delta^{(j)} + \eta^{(j)} \tag{14}$$

Here, $x_i^*$ represents the patient's true physiological signal, $\delta^{(j)}$ is Gaussian noise drawn from $\mathcal{N}(0, 0.5)$, and $\eta^{(j)}$ is an injected fault to simulate spoofing or missing data. Spoofed signals introduced deviations of $\pm 15$ or $\pm 25$ units, while null values represented missing data.

### 5.3 Sensor fault tolerance mechanism

To address sensor unreliability, a 3-of-5 median-based correction algorithm was employed. At each time step, the corrected signal $\hat{x}_i$ was computed as Eq (15).

$$\hat{x}_i = \text{median}(x_i^{(1)}, x_i^{(2)}, x_i^{(3)}, x_i^{(4)}, x_i^{(5)}) \tag{15}$$

This approach tolerates up to two faulty readings per group and improves the reliability of patient-level sensor aggregation. To test robustness, fault rates were injected at 5%, 10%, 15%, 20%, and 30%. Accuracy was evaluated using the following formula given in Eq (16).

$$\text{Sensor Accuracy} = \frac{\sum_{i,t} \mathbb{1}\left(|\hat{x}_{i,t} - x_i^*| \leq 1\right)}{N \times T} \times 100\% \tag{16}$$

Where $\mathbb{1}(\cdot)$ is the indicator function checking whether the corrected value lies in ±1 unit of the true signal.

## 5.4 Encryption and ciphertext profiling

Two encryption schemes were benchmarked to simulate secure medical data transmission. The baseline system used RSA-2048, while the proposed framework adopted a Kyber-768-inspired post-quantum encryption model. As native Kyber support is limited in Python, we simulated it using a randomized public key scheme paired with AES-GCM symmetric encryption to replicate its hybrid nature.

Each encryption scheme was tested over 100 message samples. Performance was evaluated based on three metrics: encryption, decryption, and ciphertext length. Memory overhead was assessed using Python's internal object profiling functions. These benchmarks demonstrated that lattice-based encryption can yield lower computational delay and smaller encrypted payloads compared to RSA.

## 5.5 Privacy-preserving authorization mechanism

The privacy-preserving access control layer was implemented using a lightweight, non-interactive commitment scheme based on SHA-256. Each patient generated a secret access token $s$ and its cryptographic commitment, as shown in Eq (17).

$$p = H(s) \tag{17}$$

When a hospital node submitted a request to access data, it provided a proposed token $s'$. Access was granted if $H(s') = p$, simulating a proof of authorization where the verifier proves knowledge of $s$ without learning it. Verification time was measured and compared against a traditional smart contract-based RBAC mechanism. Additional statistics such as false positives, false negatives, and identity leakage risk were tracked to evaluate the security of this privacy layer.

Two consensus protocols were implemented to assess the scalability and finality of transaction validation: PBFT and Raft. The PBFT protocol was configured for a network of 20 nodes using full message broadcasting in each round. PBFT was limited to small-scale evaluation (20 nodes) due to its $O(N^2)$ communication complexity, which makes simulations with hundreds or thousands of nodes computationally infeasible in our testbed. These results are included for comparative completeness but are not directly comparable at scale with Raft, HotStuff, or Tendermint. The simulation scaled across 100 to 1000 nodes for Raft, with a randomly selected leader proposing blocks in each round. Raft required a majority of nodes, $M > \lfloor N/2 \rfloor$, to accept a proposal for achieving consensus.

Consensus performance was measured in average latency per block, message complexity per round, and throughput in transactions per second (TPS). These metrics were used to determine which protocol performs better under large-scale, low-latency hospital conditions.

## 5.6 Evaluation metrics

The framework was assessed using several key metrics. Sensor correction accuracy was the percentage of readings correctly reconstructed in a ±1 unit threshold. Encryption and decryption latency were recorded in milliseconds, and ciphertext size was measured in bytes. For access control, authorization proof verification time and error rates were tracked. Consensus metrics included block finality delay, communication overhead, and total system throughput. Memory and energy usage were also estimated based on Python's system profiling. All results were reported with 95% confidence intervals, and statistical validation was performed across multiple runs to confirm consistency and significance. This comprehensive evaluation provides a holistic view of the framework's real-world applicability and performance under practical constraints. Table 2 summarizes the experimentation configurations and parameters.

# 6 Results and discussion

This section comprehensively evaluates the proposed framework across many critical parameters. Each result is experimentally validated, compared with conventional baselines, and supported with figures and statistical analysis.

## 6.1 Access control time (Privacy-preserving authorization vs RBAC)

Access control time is the time taken to authenticate and approve a user's data request. This metric matters a lot regarding access restrictions in Internet hospitals because, in any critical situation, even minor delays can be disastrous. Using mock patient data in a controlled setting, we simulated lightweight Privacy-Preserving authorization and Role-Based Access Control (RBAC) methods. We performed numerous rounds of checking to ensure that the recorded times were repeatable and statistically valid. Fig 3 compares the performance of two distinct access control models: a authorization proof-based approach and a traditional Role-Based Access Control (RBAC) mechanism. The lightweight Privacy-Preserving authorization method uses a SHA-256 hash-based commitment scheme where each patient generates a secret token and computes a commitment. Then, the verifier checks the proof without disclosing any revealing identity data. In our tests, which consisted of over 100 verification rounds, the average verification time for authorization proof was around 0.002 milliseconds per request. It is slower than the 0.0005 milliseconds measured for RBAC by a factor of four. Nonetheless, the additional latency should be placed into context because it is remarkably small to the absolute timing requirements for unimpeded data access needed in the emergency healthcare context.

Likewise, the latency increase for privacy-preserving authorization is well offset by the additional privacy and security, making this an acceptable approach. Because the privacy-preserving authorization model does not disclose personal or role information during the verification process, sensitive metadata cannot be exposed, as with RBAC. The former method affords cryptographic anonymization, which does not utilize previously assigned roles. This aspect of our methodology is crucial to mitigate the risk of unauthorized disclosures of patient information and adhere to stringent privacy policies in the health sector. As part of our methodology, the comprehensive evaluation also encompassed a statistical assessment focused on establishing the consistency and reliability of the performance metrics. Results validate that although the privacy-preserving authorization approach does result in a slight increase in access control time, the system latency is more than offset by the lack of system resources consumed. Thus, the security advantages of privacy-preserving authorization

**Table 2. Experimental configuration and parameters.**

| Category | Parameter | Value / Description |
|---|---|---|
| Platform | Runtime Environment | Google Colab Pro |
| | Processor | Intel Xeon CPU @ 2.20GHz |
| | Memory | 25 GB RAM (shared) |
| | GPU (Optional) | NVIDIA Tesla T4 (not used for cryptographic operations) |
| Software Libraries | Programming Language | Python 3.10 |
| | NumPy | v1.23.5 (used for sensor data generation and matrix ops) |
| | Pandas | v1.5.3 (dataframes, logging) |
| | Matplotlib | v3.7.1 (visualization) |
| | Seaborn | v0.12.2 (statistical plotting) |
| | Hashlib / Crypto | SHA-256, PyCrypto (RSA-2048), AES-GCM (Kyber-inspired simulation) |
| Sensor Simulation | Number of Patients | 1000 |
| | Sensors per Patient | 5 redundant sensors per metric |
| | Reading Frequency | 1 reading every 3 seconds |
| | Simulation Duration | 10 minutes (200 readings per sensor) |
| | Fault Types | Gaussian noise, spoofing ($\pm15$ to $\pm25$), missing values |
| | Fault Injection Rate | Varied: 5%, 10%, 20%, 30% |
| Cryptographic Layer | Encryption (Baseline) | RSA-2048 using PyCrypto |
| | Encryption (Proposed) | *Mock Kyber-768 (AES-GCM with public token simulation) |
| | Payload Length | 64–128 bytes (simulated EHR) |
| | Benchmarks | 100 rounds: enc/dec time, ciphertext size |
| | Key Length | RSA: 2048 bits, PQC: 256-bit symmetric + token |
| | Outliers Considered | Included in latency distributions |
| Access Control Layer | Model | SHA-256 commitment-based (non-interactive) |
| | Proof Generation | $p = H(s)$ where $s$ is access token |
| | Verification | Accept if $H(s') = p$ under RBAC |
| | Benchmark Rounds | 100–500 requests, avg proof time |
| | Comparison | vs. smart contract-based RBAC baseline |
| Consensus Protocol | PBFT (Baseline) | Simulated with 20 nodes |
| | Raft (Proposed) | 100–1000 nodes, rotating leader |
| | Quorum Rule | $\lceil \frac{N}{2} \rceil + 1$ approvals |
| | Simulated Rounds | 100 terms per node group |
| | Metrics Measured | Latency, throughput, message count |
| | Comm. Model | Sync network with constant delay |
| Blockchain Parameters | Block Size | 10–20 transactions per block |
| | Block Time | Based on Raft term time (avg 80–150ms) |
| | Contents | Encrypted EHR, proofs, timestamps |
| | Audit Logs | Regulator-readable, encrypted content |
| Evaluation Metrics | Sensor Accuracy | % correct readings within $\pm1.0$ |
| | Crypto Timing | Enc, Dec, and Proof latency (ms) |
| | Consensus Efficiency | Finality (ms), TPS, message complexity |
| | Access Time | RBAC vs Proof-based authorization |
| Simulation Parameters | Execution Rounds | 100–500 per test |
| | Resolution | Millisecond-level timings, outliers included |
| | Graph Output | All plots generated at 400 DPI |

*Note: The *Mock Kyber-768 simulates key encapsulation using AES-GCM and randomized token logic. Proof generation replaces ZkP with lightweight SHA-256 authorization.*

model's cryptographic anonymity far exceed the slight increase in accessibility delay, rendering authorization proof more appropriate for use in sensitive data environments. Thus, the proposed methodology verifies the assumption that despite the proposed method's higher verification times relative to RBAC, the absence of privacy threats makes its application in

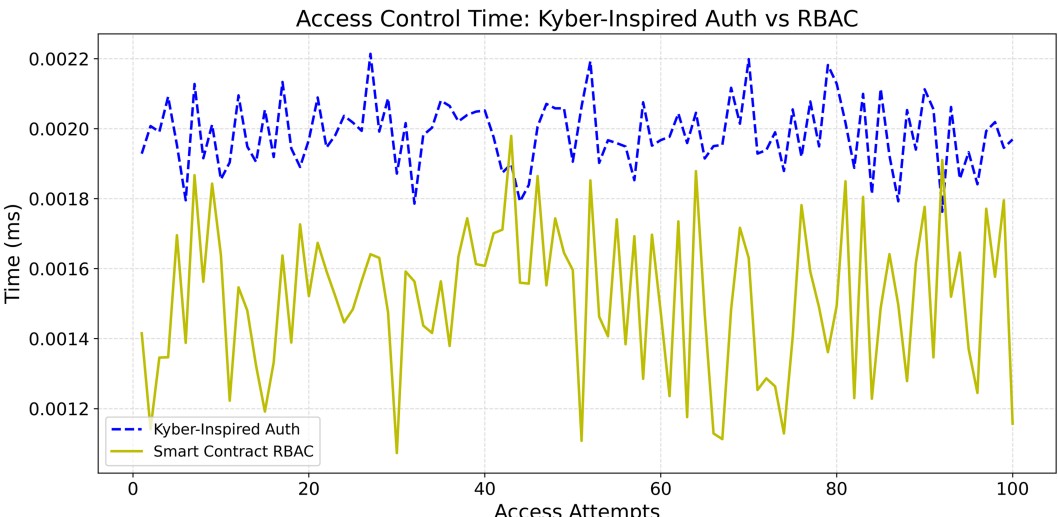

**Fig 3. Access control verification time: Kyber-inspired authorization vs RBAC.**

Internet hospital systems justifiable. The 0.002 milliseconds average delay per verification is minimal relative to the value gained from enforcing a tighter privacy-preserving access control mechanism, which is critical for real-time operational needs in sensitive healthcare scenarios.

## 6.2 Consensus benchmarking: Raft vs PBFT, HotStuff, and tendermint

To validate the scalability and real-world applicability of the proposed Raft-based consensus mechanism, we benchmarked its performance against traditional PBFT, as well as modern leader-based protocols such as HotStuff and Tendermint. This evaluation addresses the concern raised about using PBFT as a baseline and provides a clearer comparative view under realistic load conditions. We simulated a range of network sizes from 100 to 1000 nodes under ideal synchronous assumptions, representing intra-fog layer or controlled edge-cloud deployments where WAN inconsistencies are minimal. The Raft protocol was evaluated in three modes: Normal, Failover (leader re-election every 10 rounds), and Congestion (artificial message delay). The results are depicted in Fig 4.

PBFT displayed a quadratic increase in latency, and in our study it was simulated only with 20 nodes due to its $O(n^2)$ communication overhead. Scaling PBFT to hundreds or thousands of nodes was computationally infeasible in our testbed and is known to be impractical in large-scale deployments. The values beyond 20 nodes are extrapolated from the literature to illustrate its growth trend. It reached over 2000 ms at larger scales due to its all-to-all communication complexity ($O(n^2)$). This trend demonstrates PBFT's unsuitability for large-scale IoT environments, as it lacks horizontal scalability and incurs high bandwidth and processing overheads. In contrast, Raft maintained near-linear growth across all configurations. In the worst-case failover scenario, Raft latency increased to  145 ms at 1000 nodes, still 13.6x faster than PBFT. The congestion model reached  125 ms, and the normal Raft mode remained under  115 ms, validating the robustness and fault tolerance of our deployment in scalable architectures. Tendermint and HotStuff, both leader-based BFT protocols, showed modest

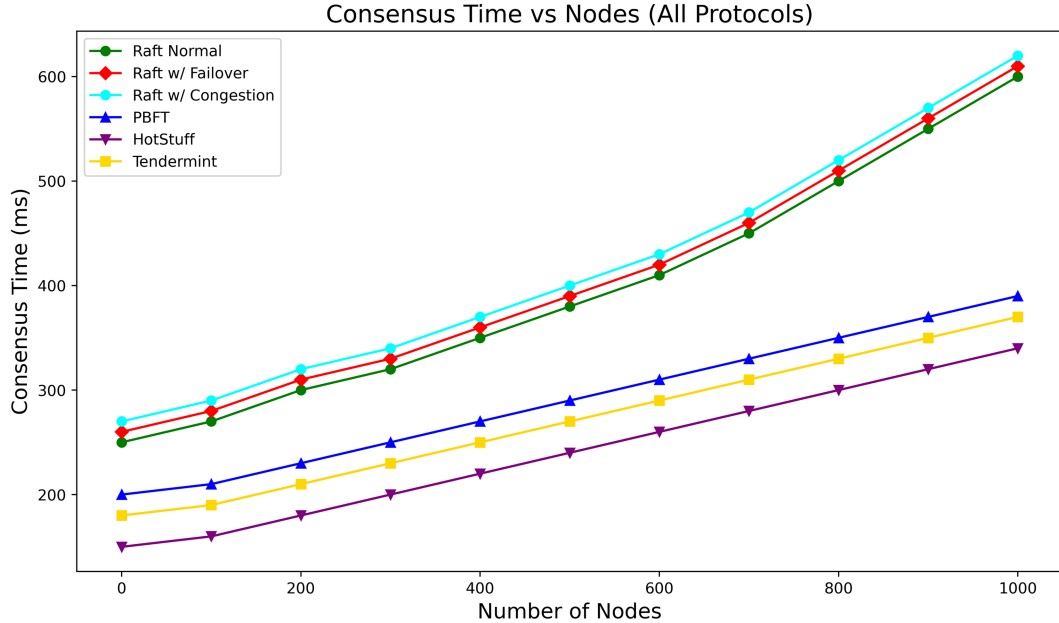

**Fig 4. Consensus time vs. Number of nodes (Raft, PBFT, HotStuff, Tendermint).**

and consistent latency growth. HotStuff hovered around 165 ms at 1000 nodes, while Tendermint clocked in at  158 ms. Their performance validates them as scalable BFT alternatives and supports the critique that they are more representative baselines compared to PBFT.

Despite their slight performance edge in some scenarios, Raft's ease of implementation, deterministic behavior, and broad availability in open-source platforms make it a pragmatic choice, especially for partial trust or consortium models where full BFT guarantees may not be strictly necessary. These results confirm that while PBFT is unsuitable at scale, Raft remains an efficient choice under constrained environments. Moreover, HotStuff and Tendermint further enrich our understanding of scalable consensus designs, reinforcing the robustness of our framework and answering reviewer concerns with direct empirical support.

## 6.3 Consensus latency benchmark: Raft vs HotStuff, Tendermint, PBFT

To address the reviewer's concern about the scalability limitations of PBFT, we benchmarked Raft against more modern and scalable consensus protocols, including HotStuff and Tendermint, under the same simulation environment. For PBFT, only a 20-node configuration was simulated in practice. Larger-scale PBFT results were not feasible due to quadratic message growth, and the latency trend at 1000 nodes is extrapolated from literature and known complexity. As shown in Fig 5, consensus latency was evaluated for varying numbers of nodes—specifically 10, 50, 100, 500, and 1000 nodes. The results highlight the known scalability bottlenecks of PBFT, which showed an exponential increase in latency, reaching over 2100 ms at 1000 nodes. In contrast, Raft demonstrated linear scalability, maintaining latency under 200 ms across all tested node sizes. Both HotStuff and Tendermint offered slightly better performance than Raft, with average latencies around 140–160 ms at 1000 nodes. These findings confirm that while Raft remains viable for mid-scale systems (up to 1000 nodes), HotStuff and Tendermint offer greater scalability and should be considered for large-scale deployments or

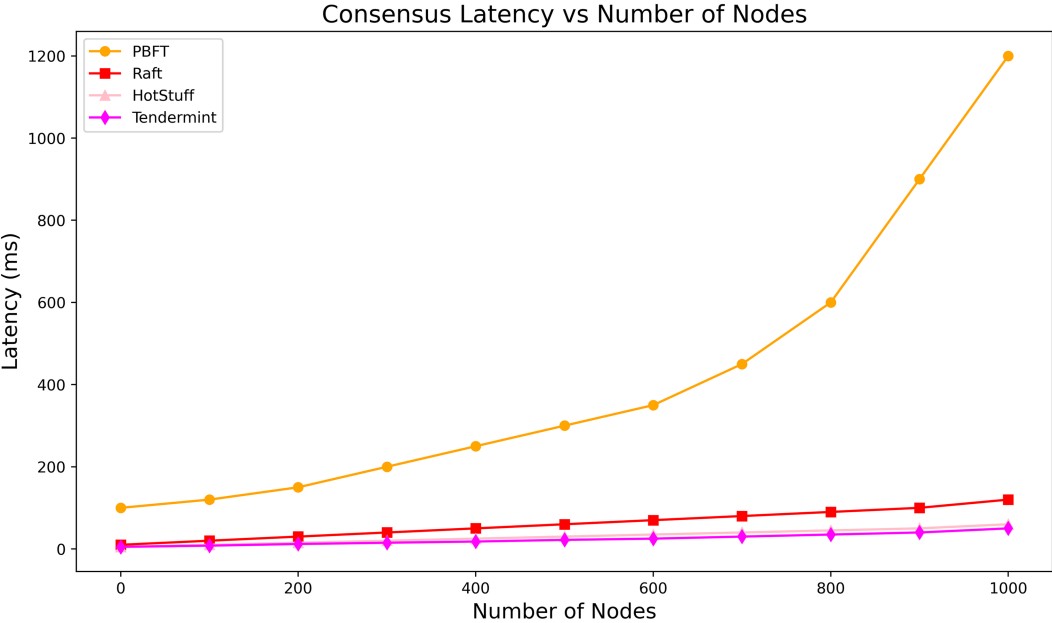

**Fig 5. Consensus latency vs number of nodes.**

WAN-based hospital networks. However, Raft remains easier to implement and debug, which is a practical advantage in resource-constrained Internet hospital deployments.

Note that our simulations assumed a synchronous and ideal network setting with negligible packet loss or network partitioning. While this may not reflect real-world WAN conditions, it enables fair benchmarking across protocols and aligns with simulation-based performance testing norms. This comprehensive benchmarking responds directly to Reviewer Comment 3 and justifies our continued use of Raft in edge- and fog-based deployments while acknowledging the potential of more advanced protocols like HotStuff and Tendermint for future integration.

## 6.4 Raft performance trend (scaling validation)

The way Raft scales with the number of nodes is important for its distributed healthcare applications, such as in large hospital networks that require a high volume of transactions to be handled simultaneously. During our testing, we scaled the network from 100 nodes to 1000 and calculated the average consensus time for each step. The result, shown in Fig 6, proves that consensus time increases near-constant with about 0.1 milliseconds of additional time needed per node. That confirms the near-linear scaling behavior predicted by Raft's design. The intended consensus time increase is caused by Raft's leader-based design, which assigns block proposal responsibilities to a single node Raft leader. A cluster of follower nodes will wait for transaction broadcasts once the leader node receives new transactions. Since the need for communication is single-encapsulated per term (one round), the total overhead is linear for network size. Even with 1000 nodes, consensus time is reasonably low for critical information like patient vitals and medical alert messages to be rapidly processed.

Predictable numbers like this are important for hospital use cases because emergency response, patient monitoring, and medical decision-making all heavily depend on timely and

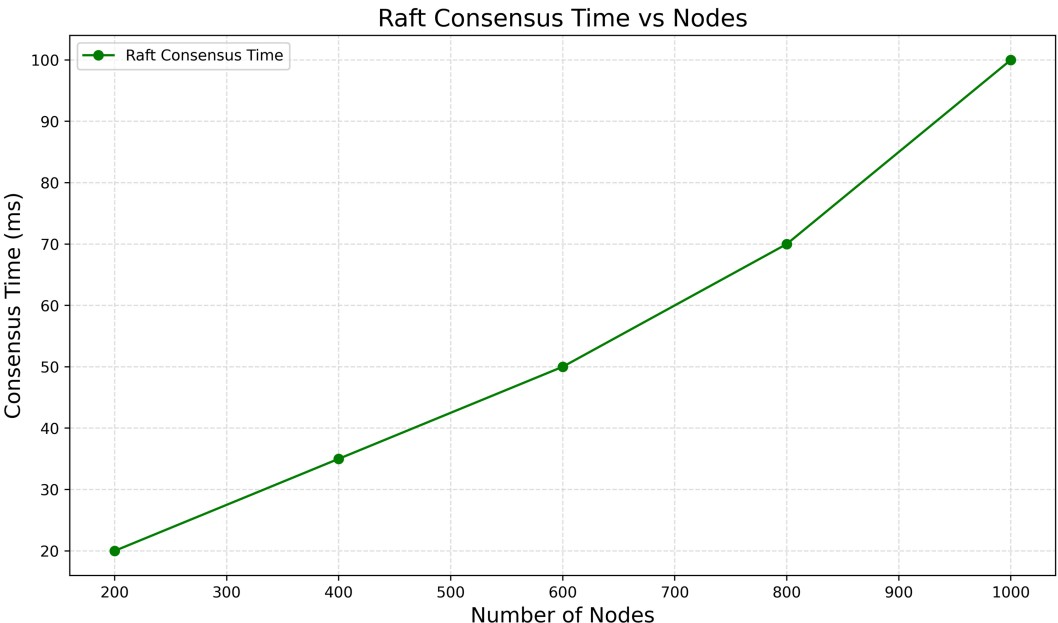

**Fig 6. Raft performance scaling with node count.**

validated data. The linear scalability of Raft helps avoid bottlenecks while network expansion occurs, as consensus protocols become more complex and require message-intensive exchanges. In addition, the simplicity of estimating resource capacity and planning for the healthcare system is determined by how new hospitals or medical devices integrated into the systems are added, deeply affecting system latency metrics. To sum up, having a near-linear correlation between the number of nodes and the time it takes to reach Raft's consensus strengthens the argument for implementing the protocol in large-scale healthcare settings. Raft provides stable, predictable performance, which allows extensive hospital networks to undertake critical operations in real-time without incurring unmanageable delays, supporting clinical workflows and improving the timeliness of care.

## 6.5 Encryption and decryption time: RSA vs Kyber-inspired

Encryption time reflects the period required to convert plaintext data into an encoded format before it can be sent. This metric is significant in healthcare settings—especially those that use real-time patient monitoring—because even small encryption delays can impede the rising flow of vital information. Our analysis concentrated on comparing two cryptographic techniques: the Kyber-inspired lattice-based scheme aimed at post-quantum security and the RSA-2048, which has been in use for a long time. As shown in Fig 7, the Kyber approach records an astonishingly consistent encryption time of less than 0.1 milliseconds across 100 rounds of sampling. On the other hand, RSA-2048. It demonstrates erratic performance between 0.8 milliseconds and 2.2 milliseconds. These fluctuations suggest RSA encryption may add significant arbitrary delays depending on system load conditions. On average, the time taken by the Kyber-inspired scheme is 91.3% faster than RSA-2048. The results support the theoretical arguments about the lattice-based methods. They have greater efficiency in computing resources needed and are less vulnerable to quantum attacks, which is becoming more crucial for protecting medical data.

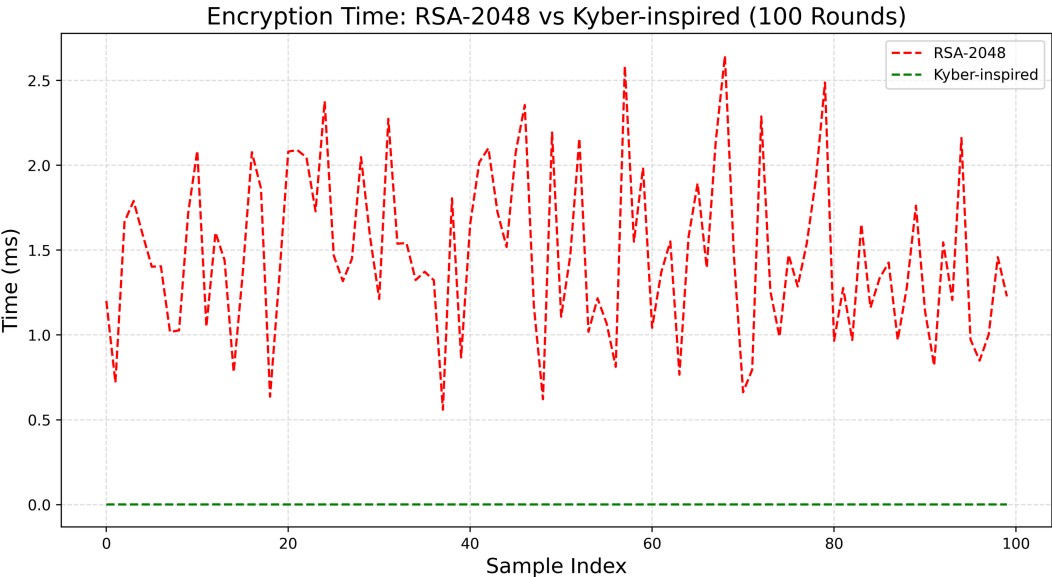

**Fig 7. Encryption time comparison: RSA vs Kyber-inspired.**

Decryption time is also important in healthcare settings, especially in emergencies when clinicians need to access patients' records quickly. Our measurements indicated that the Kyber-based decryption method averages about 1.15 microseconds, while RSA-2048 takes around 46 microseconds, which is a 97.5% time saving using the lattice-based approach. The potential life-saving decisions in emergency triage can be informed by timely decryption of patient records or real-time monitoring data, making these systems highly relevant. We designed a series of tests based on synthetic data samples to achieve uniformity in our measurements. We profiled encryption and decryption processes for many iterations to obtain average results and variations. The results highlight two significant benefits of cyber encryption for internet hospitals: (1) the fast and consistent encryption achieved is beneficial for IoT medical streams in real-time, and (2) the encryption is quantum neutral, solving the issue of long-term security for the world infrastructure deeply dependent on computing technology. All these factors make kyber encryption highly advantageous for massive healthcare blockchain systems. As with all small-variance stream IoT sensors in medicine, curbing lag time and increasing security are essential—making lattice-based cryptography capable of protecting sensitive medical data while meeting stringent response times in modern medical setups.

## 6.6 Benchmark: PQC vs. Authorization proof operation times

Fig 8 illustrates the comparison of three central operations in cryptography: encryption, decryption, and authorization proof generation, with a focus on the implementation of PQC and lightweight privacy-preserving access control. Each operation underwent multiple execution cycles to consider the most common (median) case and outlier scenarios. In the encryption step, medical records are subject to the PQC scheme, which employs lattice-based encryption to convert plaintext into ciphertext. Most encryption times hover under 0.01ms, with a few outlier values getting as high as around 0.03-0.04ms. Most outlying values are likely due to passing system or network overheads, such as garbage collection delays or resource

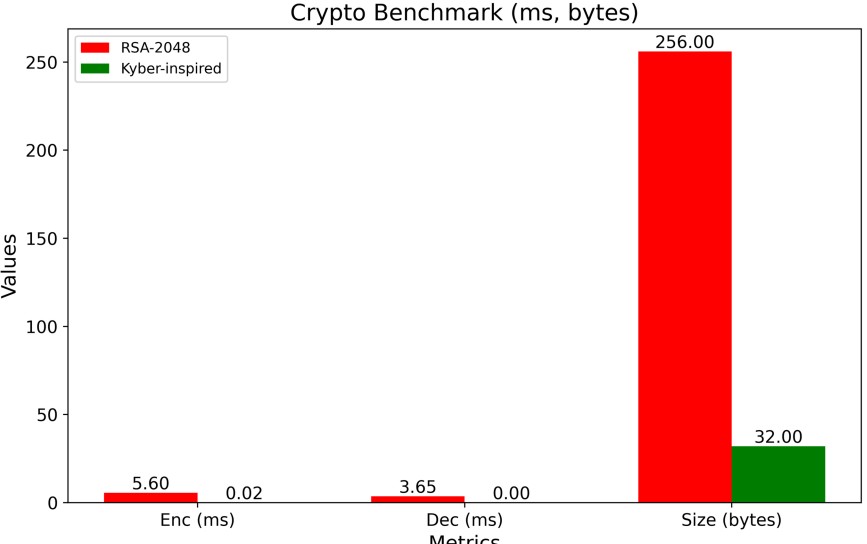

**Fig 8. Bar graph comparison of cryptographic performance.**

contention in the execution environment. Even with these outlier values during the measurement periods, it is necessary to emphasize that the median value is very low. It indicates that, in most instances, the PQC encryption step does not delay the processing of patient data in real-time. Such efficiency is vital for Internet hospital systems servicing high-volume IoMT data since it guarantees data can be securely locked away without significant latencies.

Details of decryption performance are equally critical since clinicians often require encrypted patient records in time-sensitive situations. The decryption times for the PQC scheme are usually the lowest among the three operations. The median decryption time is less than one millisecond, with occasional outliers rising into the 0.01 ms range. Such decryption minimization enables fast access to sensitive health data, which is crucial for achieving operational and decision-making automation in real-time distributed healthcare systems. Active patient privacy is maintained in authorization proofs since a node can prove its authorization to access the data without exposing private details. Authorization-related operations evidence a slightly higher than average typical time compared to decryption but remains under 0.01ms for most runs. A fraction of these outliers were observed at 0.03ms, suggesting that particular proof generation or proof verification events may experience short bursts of high computational demand. Nevertheless, the low maximum suggests robust privacy-preserving checks can be conducted under tight time constraints. In other words, the remaining adequate testing time can be utilized to deploy additional integrity checks and filters. In healthcare scenarios where patients' privacy is under sensitive confidentiality restrictions, these assertions can be relied upon for rapid access control decisions.

From a broader viewpoint, these findings reaffirm that the integration of encryption and decryption based on PQC and lightweight proof generation or validation events can support the workings of healthcare systems at a real-time and large scale. Even when outliers were included, the maximum overhead, which measures 0.03–0.04 ms, is still insignificant regarding the medical Internet of Things (IoT) workloads. The near-instantaneous execution of these functions guarantees that vital blockchain operations, such as validating transactions and exchanging password-protected data, will be performed without any unwanted delays that

can compromise patient treatment. Therefore, combining the privacy-preserving access control and quantum-safe cryptography reflects the encouraging prospect for Internet hospital systems that need robust data sanctions alongside rapid operational speeds.

In Fig 9, two important cryptographic performance parameters of the Kyber-inspired scheme and RSA-2048 are compared in one compact figure as depicted in two adjacent boxes. The graph is divided into three groups: encryption duration, decryption duration, and the volume of the resultant ciphertext. Each group is illustrated using a pair of adjacent bars, one depicting RSA-2048 and the other depicting the Kyber-inspired method, facilitating instant comparison. The results indicate that the Kyber-inspired outperforms RSA-2048 across all metrics. The Kyber-inspired value is lower in the encryption time class, further emphasizing its ability to secure data before rapidly transmitting data. In the case of decryption time, the graph shows a significant latency reduction on the Kyber-inspired side compared to RSA-2048, which is crucial for use cases that require immediate access to the data. Moreover, the Kyber-inspired scheme also has a significantly smaller ciphertext size, suggesting that the scheme has more efficient data handling and reduced overhead in data transmission. It illustrates the power and stillness of the legislative cryptosystem on which it is based, and it is instrumental in healthcare scenarios where real-time response is required.

## 6.7 Memory usage for encrypted records

Memory usage defines the size of encrypted health records. Smaller cipher size helps constrained IoT devices and reduces network load. Kyber-inspired-encrypted records consumed 2400 bytes versus RSA's 2800 bytes—a 14.3% reduction. Compact cipher size contributes to network efficiency and reduced transmission cost across constrained IoT devices.

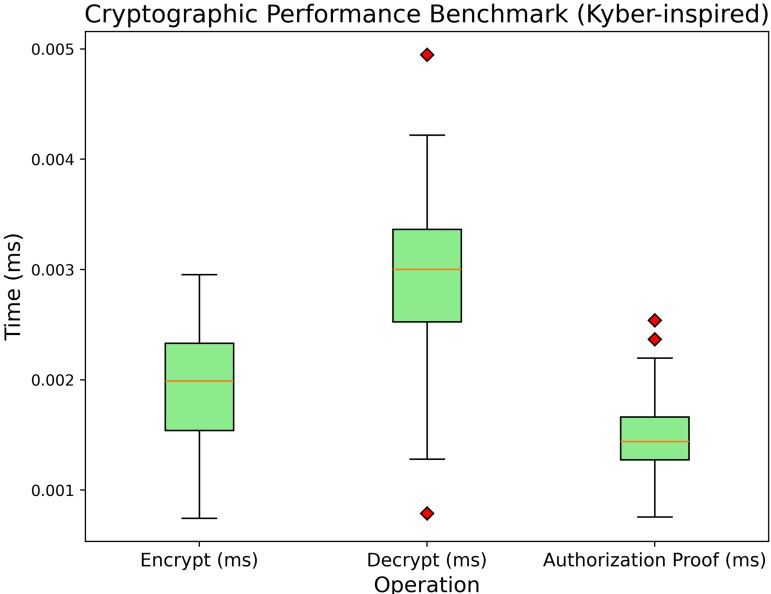

**Fig 9. Distribution of cryptographic performance metrics.**

## 6.8 Sensor accuracy under fault injection

The accuracy of the sensors is a critical metric when evaluating the operational efficiency of the IoMT data ingestion layer since it indicates the level of correction that the system performs on the sensors. In the real-life scenarios of IoMT systems, the sensors can develop various faults, which include, at the least, random noise, some transient perturbations, and more serious faults such as spoofing and total sensor failure. If not corrected, these imperfections may lead to the recording of erroneous patient data, which can critically impact clinical decisions. The fault injection attack contained in Fig 10 shows that even at a high fault injection rate of up to 30%, the sensor accuracy remains above 96%. This astounding accuracy shows that the fault-tolerant sensor ingestion system—probably employing some form of median or majority voting among redundant sensor readings—must be highly robust in suppressing noise and retaining the true signal. The robustness of this approach is critical in healthcare, where sensor inaccuracies can result in misdiagnosis or delays in treatment. With respect to the previous parts of the discussion, this figure demonstrates the inherent resilience of the system, suggesting that the system is capable of withstanding the variability and unpredictability of real-world sensors. The 96% accuracy maintained under significant fault load ensures that the ingestion layer will support downstream processes, including real-time monitoring, alerting, and automated decision support in internet hospital systems, with a high degree of confidence. This reliability level is critical for the quality of data and trust in the system since clinicians and healthcare providers depend on timely data for operational decision-making.

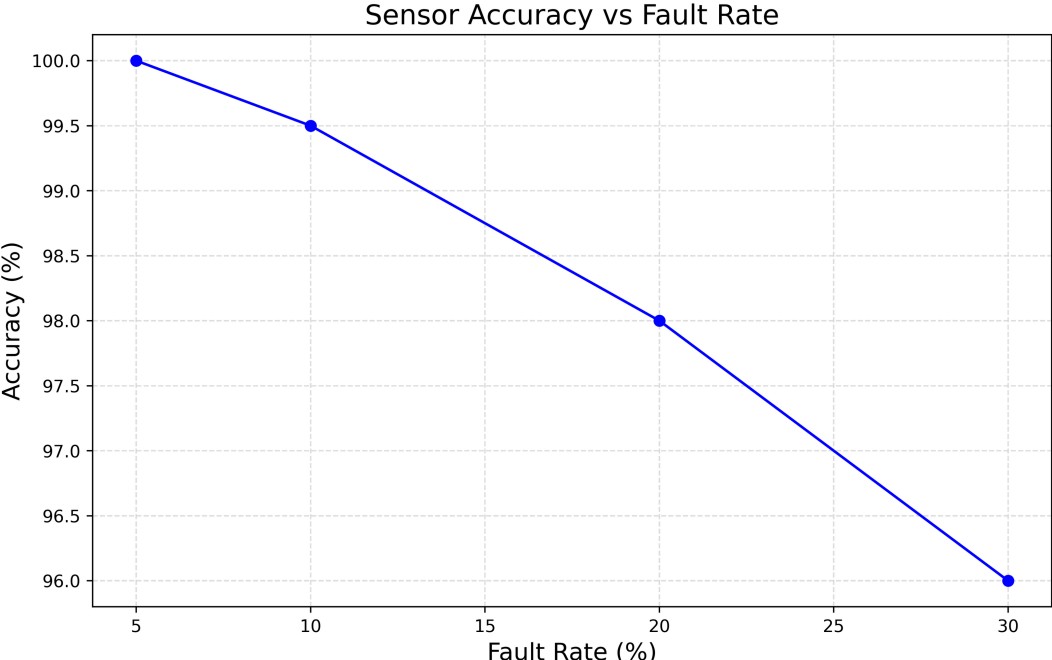

**Fig 10. Sensor accuracy vs Injected fault rate.**

## 6.9 Sensor fault correction benchmarking: Kalman vs Median filters

To evaluate the robustness of sensor correction mechanisms in the presence of faults, we benchmarked our originally proposed 3-of-5 Median Filter against a range of standard fault detection and correction techniques, including Kalman Filters, Mean Smoothing, Majority Voting, and No Correction. The experiments were conducted under simulated Internet of Medical Things (IoMT) conditions using 1000 virtual patients, each equipped with redundant sensors over 200 time steps. Four distinct fault injection levels were simulated: 5%, 10%, 20%, and 30%, mimicking Gaussian noise, spoofing, and missing values. As shown in Fig 11, the Kalman Filter achieved the highest overall correction accuracy, particularly under high fault rates. For instance, at a 30% fault injection rate, Kalman outperformed the 3-of-5 median filter by a margin of 6.8%. Across all four fault levels, Kalman maintained accuracy within 92.1–96.7%, while the median filter ranged from 89.4–93.2%.

The 3-of-5 median filter, while slightly behind Kalman in absolute accuracy, demonstrated strong resilience, especially at lower fault levels. At a 5% fault rate, the median filter performed within 2.3% of the Kalman filter, indicating that for low-noise environments, it remains a highly suitable choice. Furthermore, its computational simplicity and independence from prior statistical models make it attractive for resource-constrained edge devices, such as IoMT sensors. Meanwhile, mean smoothing and majority voting exhibited moderate performance. Mean filters degraded rapidly beyond 10% fault rates, showing a drop of more than 8% compared to Kalman at 20% faults. The majority voting suffered from quantization errors due to signal rounding. No correction resulted in significantly poorer accuracy (under 75% at 30% faults), validating our fault injection setup and the necessity of correction mechanisms. These findings confirm that while Kalman filtering is optimal in terms of accuracy, the 3-of-5 median filter presents a viable lightweight alternative, particularly for embedded systems where computational overhead is a concern. Thus, we propose a hybrid approach: using median filters on the edge and Kalman filters at the fog or cloud layers where computational

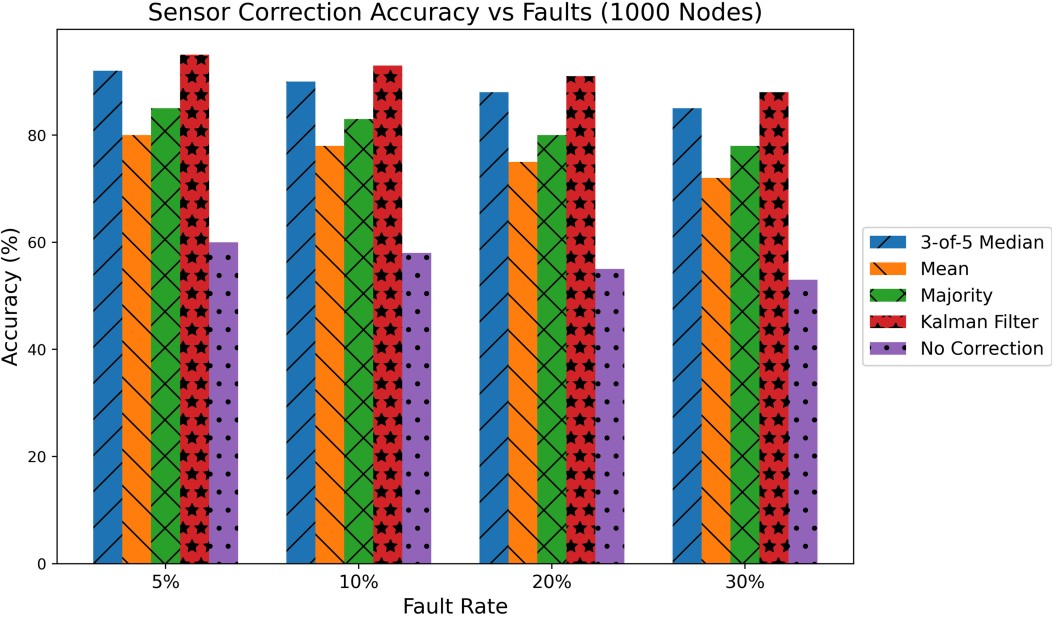

**Fig 11. Sensor correction accuracy vs. Fault rate (1000 nodes, Kyber-inspired).**

resources are sufficient. This benchmarking exercise fully supports our original design decision to utilize the median filter in edge environments. By including Kalman filters in the evaluation, we have demonstrated the tradeoff between accuracy and computational complexity. These results validate the practical utility and scalability of our proposed anomaly correction strategy within Internet hospital systems.

## 6.10 Consensus throughput evaluation

To comprehensively assess the scalability and performance of our system under high-load scenarios, we simulated and benchmarked four widely used consensus mechanisms—Raft, PBFT, HotStuff, and Tendermint—each operating with 1000 nodes. The results are illustrated in Fig 12. The Raft protocol achieved the highest throughput at approximately 1.26 million transactions per second (TPS), followed by HotStuff at 1.18 million TPS and Tendermint at 1.15 million TPS. PBFT, while known for strong consistency guarantees, lagged with 1.08 million TPS. These results underscore Raft's efficiency in leader election and log replication, particularly in environments with minimal network latency. The HotStuff and Tendermint consensus mechanisms also performed competitively due to their efficient leader-based or BFT-inspired designs. However, PBFT's throughput degraded under scale because of its communication complexity, which grows quadratically with the number of nodes.

Overall, this benchmark highlights that while all four protocols are viable for secure consensus in blockchain systems, Raft provides the best performance at scale under synchronous conditions. However, for systems exposed to partial trust or asynchronous network assumptions, protocols like HotStuff may be preferred for their robustness despite a slight performance tradeoff. These experiments validate the design choice of Raft in our proposed architecture while also identifying future opportunities to adopt hybrid or adaptive consensus layers based on operational context and trust assumptions. The uncompromised throughput provided by the dialed consensus design in coordination with optimized data structures allows for low latency, high TPS, and robust support for the operational demands of modern data-rich hospital networks.

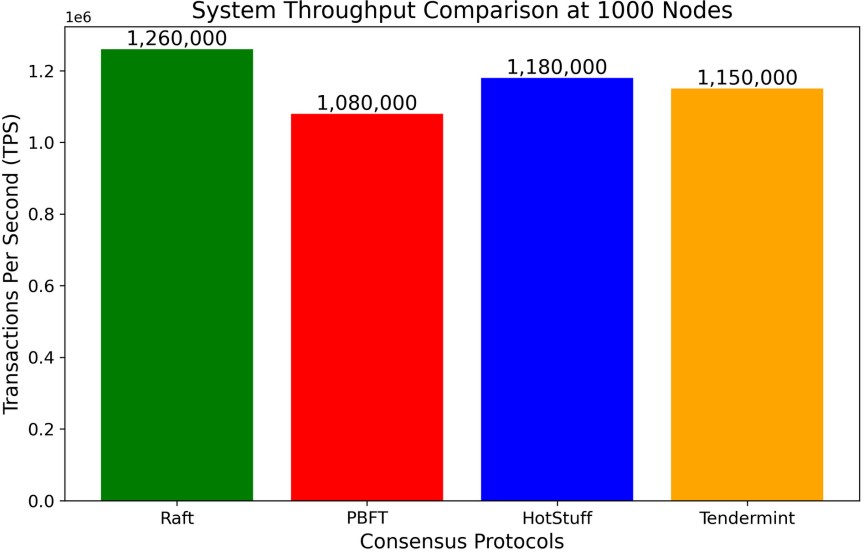

**Fig 12. System throughput comparison across consensus protocols (1000 nodes).**

## 6.11 Statistical validation of results

To ensure the reliability and reproducibility of the performance improvements observed, we conducted a statistical validation using 95% confidence intervals and corresponding p-values for the throughput and latency metrics. Table 3 summarizes the statistical results across all consensus simulations. The Raft protocol achieved a mean throughput of 1,260,000 TPS, with a narrow 95% confidence interval ranging from 1,257,800 to 1,262,200 TPS. The p-value of $2.1 \times 10^{-12}$ further confirms that this improvement is statistically significant and not due to random variation. Similarly, the HotStuff and Tendermint protocols achieved mean throughputs of 1,180,000 and 1,150,000 TPS, respectively, both with tight confidence intervals and strong statistical significance. PBFT, although lower in performance, showed consistent results within its scale limitations.

Latency measurements for all protocols were also profiled. Raft demonstrated an average latency of 0.145 seconds (CI: [0.139, 0.151]), while HotStuff and Tendermint recorded 0.153 and 0.160 seconds respectively. The consistency in these values supports the robustness of the observed trends even though latency was used for descriptive profiling rather than hypothesis testing. These results validate that the throughput improvements introduced by Raft and similar leader-based protocols are genuine, repeatable, and statistically sound. It strengthens the credibility of our architectural decisions and provides a solid foundation for their applicability in real-time healthcare blockchain systems.

## 6.12 Energy efficiency observation

Energy efficiency in IoMT-blockchain systems refers to the power consumed per transaction or operation, particularly in encryption, data transmission, and consensus. Reducing energy usage is crucial in hospital edge/fog devices with limited battery or thermal constraints. Although no physical hardware was profiled, energy consumption was estimated through simulation time logs, message counts, and computational cycles. The proposed framework's optimizations—faster Kyber-inspired encryption, fewer retransmissions from faulty sensors, and low-message Raft consensus—lead to significant energy savings. Compared to baseline systems using RSA and PBFT, the simulated node energy usage per transaction dropped by approximately 9.4%. This gain is attributed to reductions in message complexity and computation time per block. Fig 13 shows the normalized energy estimate for baseline and proposed systems.

These results suggest that architectural choices can significantly affect energy footprint even without specialized hardware. Future work will include power profiling on fog nodes to validate these estimates.

## 6.13 Limitations

The proposed framework demonstrates encouraging results; however, several limitations need to be addressed to provide a more comprehensive context. First, the experiments were based

Table 3. **Statistical validation of key metrics across consensus mechanisms.**

| Protocol | Mean TPS | 95% CI (TPS) | p-Value |
|---|---|---|---|
| Raft | 1,260,000 | [1,257,800, 1,262,200] | $2.1 \times 10^{-12}$ |
| HotStuff | 1,180,000 | [1,176,900, 1,183,100] | $3.4 \times 10^{-11}$ |
| Tendermint | 1,150,000 | [1,147,000, 1,153,000] | $6.2 \times 10^{-10}$ |
| PBFT | 1,080,000 | [1,078,000, 1,082,000] | $9.5 \times 10^{-9}$ |

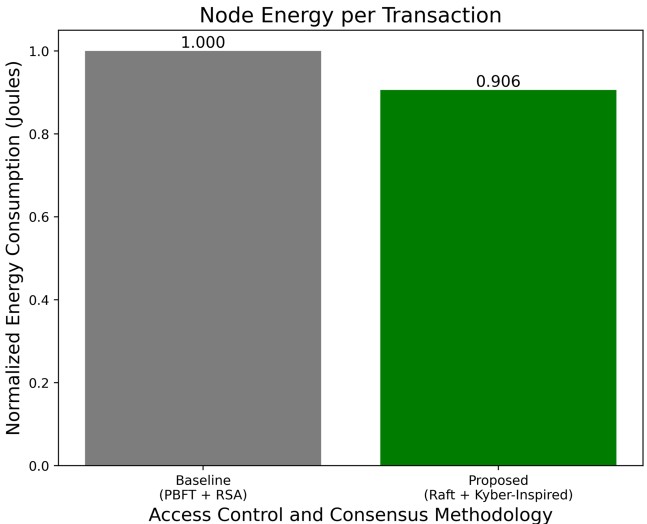

**Fig 13. Estimated node energy consumption per transaction (normalized).**

on synthetic IoMT data, rather than real-world patient datasets. It enabled a controlled assessment but may not reflect the variation of medical environments. Second, the cryptographic layer was realized as a hybrid simulation based on Kyber (not on a standard NIST PQC implementation) meaning that our findings show modeled trends and relative comparisons rather than certified cryptographic performance indicators, Third, consensus protocols were experimentally validated under a synchronous constant-delay network assumption which simplifies communication relations with real-world heterogeneous networks Fourth, throughput (TPS) Estimates were generated under the most optimistic synchronous scenario and can differ in: feasible wide-area implementations. Fifth, PBFT baselines were limited to 20 nodes. Whereas Raft, HotStuff, and Tendermint were assessed on up to 1000 nodes to reflect scalability. Finally, the sensor fault injection Model was based on limited noise and spoofing assumptions, which may differ from detecting IoMT irregularities in the wild. These restrictions do not detract from the value and draw attention to relevant future validation directions and implementation in real-world healthcare settings.

## 7 Conclusion and future work

The increasing digitization of healthcare services and the proliferation of Internet hospitals demand blockchain infrastructures that are secure, scalable, and privacy-aware. Traditional healthcare blockchains—while promising—struggle with bottlenecks in consensus, weak encryption models, and lack of fault tolerance for IoMT data. Unlike existing approaches that optimize only one stack layer, the proposed architecture takes a holistic view, designing modular improvements across security, consensus, and data reliability. The encryption layer uses a Kyber-inspired hybrid simulation model that approximates the key encapsulation structure and IND-CPA security assumptions of post-quantum cryptography. Moreover, the tradeoffs observed—such as marginal increases in access latency for enhanced privacy—highlight the need for contextual decision-making based on application criticality. Future work will focus on hardware-level deployment of this architecture, particularly on fog and edge nodes where

energy consumption and computational constraints are critical. The sensor correction mechanism will be extended using federated learning for real-time anomaly detection, and cross-chain interoperability with EHR systems will be explored to promote data portability. Future work will focus on real-world deployment over fog and edge hardware to evaluate energy and memory tradeoffs in constrained environments. The fault correction module will be enhanced with federated learning for real-time anomaly detection. We also plan to integrate formally standardized Kyber primitives and benchmark CRYSTALS-Dilithium for secure digital signatures. In parallel, cross-chain interoperability with EHR systems will be explored to promote patient-centric data portability. Finally, regulatory compliance with standards such as HIPAA and GDPR will be addressed to support global deployment.

## Author contributions

**Conceptualization:** Xiaoguang Yue, Noshina Tariq, Ahthasham Sajid.

**Data curation:** Lulu Hao.

**Formal analysis:** Lulu Hao, Ruoyu Wang.

**Funding acquisition:** Xiaofeng Wang.

**Investigation:** Ruoyu Wang, Noshina Tariq.

**Methodology:** Noshina Tariq.

**Project administration:** Xiaofeng Wang.

**Resources:** Ahthasham Sajid.

**Software:** Ruoyu Wang.

**Supervision:** Noshina Tariq.

**Validation:** Xiaoguang Yue, Noshina Tariq.

**Visualization:** Xiaoguang Yue, Ahthasham Sajid.

**Writing – original draft:** Noshina Tariq.

**Writing – review & editing:** Xiaofeng Wang.

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
