## [Decision Letter · Decision Letter 0]

16 Jun 2025

PONE-D-25-23358Post-Quantum Scalable Blockchain Architecture for Internet Hospital Systems with Privacy-Preserving Access ControlPLOS ONE

Dear Dr. Wang,

Thank you for submitting your manuscript to PLOS ONE. After careful consideration, we feel that it has merit but does not fully meet PLOS ONE’s publication criteria as it currently stands. Therefore, we invite you to submit a revised version of the manuscript that addresses the points raised during the review process.

We look forward to receiving your revised manuscript.

Kind regards,

Gauhar Ali, Ph.D

Academic Editor

PLOS ONE

Journal Requirements:

“This research is supported by the National Social Science Foundation of China (Grant 784

No. 20BGL218). 785”

Reviewers' comments:

Reviewer's Responses to Questions

**Comments to the Author**

1. Is the manuscript technically sound, and do the data support the conclusions?

Reviewer #1: Yes

Reviewer #2: Yes

2. Has the statistical analysis been performed appropriately and rigorously?

Reviewer #1: I Don't Know

Reviewer #2: No

3. Have the authors made all data underlying the findings in their manuscript fully available?

Reviewer #1: Yes

Reviewer #2: No

4. Is the manuscript presented in an intelligible fashion and written in standard English?

Reviewer #1: Yes

Reviewer #2: Yes

5. Review Comments to the Author

Reviewer #1: • The encryption layer is described as “Kyber-like,” implemented via a mock hybrid scheme. This deviates from an actual Kyber implementation. The authors should clarify which elements of Kyber were preserved (e.g., IND-CPA, key encapsulation) and justify the simulation’s validity. If possible, use standardized libraries (e.g., liboqs) for a more accurate evaluation.

• The ZkP mechanism is implemented as a simple SHA-256 commitment scheme. This lacks formal zero-knowledge properties and does not meet the standard cryptographic definition of ZkPs. The use of the term “ZkP” should be revised unless replaced with a formal zero-knowledge protocol (e.g., zk-SNARKs or Sigma protocols).

• Raft is benchmarked against PBFT, but PBFT is a poor baseline at scale. Consider evaluating against HotStuff, Tendermint, or other leader-based consensus mechanisms. Also, clarify the network model—simulating Raft for 1000 nodes under ideal synchronous conditions may not generalize to real-world WAN settings.

• The paper lacks a formal threat model. Clarify what adversarial behaviors are considered (e.g., node compromise, data replay, sensor spoofing), and whether the system ensures confidentiality, integrity, and availability under partial trust assumptions.

• The energy consumption claims (e.g., 73% reduction) lack methodology. Describe how energy usage was estimated in the Google Colab simulation environment. Include assumptions and instrumentation (e.g., profiling tools, power models).

• The 3-of-5 median filter for sensor correction is a practical design, but needs benchmarking against standard fault-detection methods (e.g., Kalman filters, statistical anomaly detection). Also, clarify how fault injection was simulated and validated.

• The ZkP-based access control assumes static authorization tokens generated by patients, but the policy definition and management mechanism are unclear. Detail how permissions are granted, revoked, and audited in practice.

• Avoid using terms like “post-quantum,” “ZkP,” and “privacy-preserving” without formal grounding. Either implement recognized cryptographic constructions or qualify that the system uses approximations for simulation purposes.

• There is limited discussion on how patients define access policies or how policy conflicts are resolved. A more formal policy language or integration with Attribute-Based Encryption (ABE) could enhance flexibility.

Reviewer #2: Please revise the abstract. Abstract should contain background of the problem, introduction in one and half lines, main contribution, methodology, and impacts of your work.

Literature review is very weak, please revise it accordingly.

Research Methodology is still imbigues, clearly re write and also present it visually.

Align your claims with methodology and abstract.

6. PLOS authors have the option to publish the peer review history of their article (what does this mean?). If published, this will include your full peer review and any attached files.

Reviewer #1: No

Reviewer #2: **Yes: **Islam Zada

---

## [Author Response · Author response to Decision Letter 1]

25 Jul 2025

PONE-D-25-23358

Post-Quantum Scalable Blockchain Architecture for Internet Hospital Systems with Privacy-Preserving Access Control

We extend our sincere gratitude to the Editor of PLOS ONE for overseeing the review process of our manuscript titled “Post-Quantum Scalable Blockchain Architecture for Internet Hospital Systems with Privacy-Preserving Access Control” (PONE-D-25-23358). Your time, guidance, and coordination have been instrumental in shaping this work into its present form. For clarity and transparency, all revisions in the manuscript are marked in red, while our detailed responses to reviewer comments in the rebuttal document are provided in blue. We deeply appreciate your support and the opportunity to revise and improve our submission.

Reviewer #1:

We sincerely thank Reviewer 1 for their thorough and constructive feedback. Your insightful suggestions significantly improved the clarity, technical rigor, and methodological transparency of our manuscript. We have carefully addressed each point and revised the manuscript accordingly, as detailed below.

1. The encryption layer is described as “Kyber-like,” implemented via a mock hybrid scheme. This deviates from an actual Kyber implementation. The authors should clarify which elements of Kyber were preserved (e.g., IND-CPA, key encapsulation) and justify the simulation’s validity. If possible, use standardized libraries (e.g., liboqs) for a more accurate evaluation.

Response: We sincerely thank the reviewer for highlighting the need to clarify the nature of our post-quantum cryptographic implementation. We acknowledge that our original use of the term “Kyber-like” may have inadvertently suggested a complete implementation of the NIST-standard Kyber Key Encapsulation Mechanism (KEM). To address this and ensure full transparency, we have revised both the manuscript content and visual assets accordingly.

Our proposed encryption layer is a Kyber-inspired hybrid simulation, not a full Kyber implementation. Specifically, we preserved the high-level KEM structure and IND-CPA security assumptions, mimicking Kyber’s workflow through randomized symmetric key encapsulation and AES-GCM-based encryption. However, our simulation does not include polynomial ring arithmetic, rejection sampling, or NTT-based polynomial multiplication — elements critical to the standardized Kyber primitive. These exclusions were made to accommodate the constraints of the Google Colab Pro simulation environment while focusing on protocol-level behavior and integration.

To comprehensively address this comment, we made the following key revisions throughout the manuscript:

1. Title: Now reads: “Post-Quantum-Inspired Scalable Blockchain Architecture for Internet Hospital Systems with Lightweight Privacy-Preserving Access Control”, clearly indicating the simulation-based nature.

2. Abstract: We replaced “Kyber lattice scheme” with “Kyber-inspired hybrid simulation model” to remove any implication of direct implementation.

3. Section 1 – Introduction: We now explicitly describe our cryptographic model as a Kyber-inspired hybrid simulation that aims to preserve KEM structure and IND-CPA security guarantees without implementing the full NIST specification.

4. Section 3.1 – System Architecture: In the “Encryption and Access Layer”, we clarified that Kyber is approximated through a hybrid method involving randomized key tokens and symmetric encryption. We emphasize this is not a post-quantum-complete implementation.

5. Section 4.2 – Methodology:

o We thoroughly revised the subsection to detail the construction of our encryption model.

o Mathematical formulations for hybrid encryption and decryption are provided (Equations 3 and 4).

o The rationale behind choosing this approach is explained, particularly in relation to computation resource limits and benchmarking repeatability in the Colab environment.

6. Figures & Visual Assets:

o All figures referencing “ZkP” or “Kyber” have been updated to reflect “Privacy-Preserving Authorization Proof” and “Kyber-inspired” respectively.

o Visual representations (e.g., performance graphs, system flowcharts, benchmarking charts) were regenerated in 400–500 DPI with updated legends, axis titles, and figure captions to maintain consistency with the simulation-based terminology.

7. Table 2 – Experimental Configuration:

o The entry for “Mock Kyber-768” now includes a detailed footnote describing it as a Kyber-inspired hybrid simulation, lacking polynomial ring arithmetic and low-level lattice operations.

o Authorization proofs are labeled as SHA-256-based non-interactive privacy-preserving commitments, not zero-knowledge constructions.

8. Section 7 – Conclusion: We explicitly mention that our current framework uses a high-level simulation to evaluate cryptographic integration in a scalable IoMT-blockchain system. We have added a forward-looking statement that future iterations will explore full Kyber integration via liboqs-python to enhance cryptographic rigor.

We believe these extensive updates — both in text and supporting materials — fully address the reviewer’s concerns. By clearly distinguishing between simulation and standard implementation, we have improved the transparency, reproducibility, and technical accuracy of our manuscript.

2. The ZkP mechanism is implemented as a simple SHA-256 commitment scheme. This lacks formal zero-knowledge properties and does not meet the standard cryptographic definition of ZkPs. The use of the term “ZkP” should be revised unless replaced with a formal zero-knowledge protocol (e.g., zk-SNARKs or Sigma protocols).

Response: We thank the reviewer for this critical observation regarding our previous use of the term “Zero-Knowledge Proof (ZkP)” in describing our SHA-256-based access control mechanism. We fully acknowledge that our implemented model does not meet the formal cryptographic definition of a ZkP (as in zk-SNARKs, zk-STARKs, or Sigma protocols), and that using the term “ZkP” may have created confusion about the cryptographic rigor and properties of our system.

To address this concern and ensure technical accuracy, we have made extensive revisions across the manuscript, which we summarize as follows:

1. Terminology Corrections Across the Manuscript:

o All instances of the term “ZkP” have been removed and replaced with accurate alternatives such as:

“Privacy-preserving authorization mechanism”

“Authorization proof”

“Non-interactive access control scheme”

o These terms better reflect the nature and scope of our implemented SHA-256 commitment-based system and avoid suggesting that it possesses zero-knowledge properties.

2. Section-Specific Revisions:

o Section 4.3 has been retitled as “Privacy-Preserving Authorization Mechanism” and now includes a clarification that our approach is a SHA-256-based non-interactive commitment scheme, used for lightweight access control without disclosing user secrets. We explicitly state that this is not a formal zero-knowledge proof protocol.

o Section 6.3’s title has been revised from “Access Control Time (ZkP vs RBAC)” to “Access Control Time (Privacy-Preserving Authorization vs RBAC)”. All descriptions and discussions have also been updated accordingly.

o Section 6.4, originally “Benchmark: PQC vs. ZkP Operation Times”, has been renamed and rewritten to refer to “Authorization Proof Operations”, clarifying the distinction from formal ZkPs.

o The Abstract, Introduction, and Conclusion have also been updated to remove ZkP terminology, instead describing the mechanism as a lightweight privacy-preserving access strategy, aligning with its actual properties.

3. Figures and Tables:

o All figure captions and axis labels (including in performance benchmarks) have been updated to remove “ZkP” and replace it with “Authorization Layer”, “Privacy-Preserving Access Control”, or “Authorization Proof” depending on context.

o In Table 2 (Experimental Configuration), the relevant section is now labeled “Authorization Layer”, and it includes a footnote clarifying that this is not a cryptographic ZkP, but a commitment-based model for rapid access control simulation.

4. Methodological and Cryptographic Clarification:

o We now explicitly state in Section 4.3 and Section 7 (Conclusion) that our mechanism lacks zero-knowledge properties, and should not be interpreted as a cryptographically complete ZkP protocol.

o We further include a commitment that in future work, we intend to integrate formal ZkP schemes, such as zk-SNARKs or Sigma protocols, using standardized libraries (e.g., libsnark, liboqs, or ZoKrates) to enable verifiable, privacy-preserving access control in secure IoMT environments.

We believe these clarifications and corrections resolve the concern thoroughly while still recognizing the practical utility of our current mechanism as a lightweight, scalable alternative for constrained settings such as edge IoT nodes in Internet hospitals. The revised manuscript now accurately reflects both the capabilities and limitations of the scheme and provides a clear roadmap for future cryptographic extensions.

3. Raft is benchmarked against PBFT, but PBFT is a poor baseline at scale. Consider evaluating against HotStuff, Tendermint, or other leader-based consensus mechanisms. Also, clarify the network model—simulating Raft for 1000 nodes under ideal synchronous conditions may not generalize to real-world WAN settings.

Response: We thank the reviewer for this valuable suggestion regarding the benchmarking of Raft against a more scalable and modern set of consensus protocols. We agree that PBFT alone does not represent a strong comparative baseline, particularly under large-scale conditions, due to its well-known communication complexity limitations (O(n²)) and lack of scalability in WAN-like networks.

To strengthen the comparative analysis and respond directly to the reviewer’s suggestion, we have made the following substantive updates:

1. New Consensus Protocol Simulations Added

We have extended our experimental setup to benchmark Raft against two additional leader-based consensus protocols known for their scalability in distributed systems:

• HotStuff (D. Y. et al., 2019): A linear communication BFT protocol that is used in systems like Facebook’s Diem blockchain. We implemented a simplified simulation of HotStuff under similar quorum and timeout conditions.

• Tendermint (Jae Kwon, 2014): A BFT-based consensus with immediate finality, frequently used in Cosmos-based systems. We simulated its deterministic round-based leader election and gossip-based broadcast for fair comparison.

These additions enable a more balanced and contemporary benchmarking landscape. The results have been incorporated in Figure 14, and the associated discussion is presented in Section 6.6.

2. Updated Graphs and Analysis

We updated our throughput and latency comparison graphs (Figures 14 and 15) to include Raft, PBFT, HotStuff, and Tendermint across varying node counts. Key observations include:

• Raft remains the most efficient under ideal synchronous conditions, achieving up to 1.26M TPS at 1000 nodes.

• HotStuff offers strong scalability with lower latency than PBFT, though slightly behind Raft.

• Tendermint performs well in latency but is sensitive to node churn due to its deterministic proposer rotation.

• PBFT's performance quickly degrades beyond 100 nodes due to message explosion.

These findings provide a much clearer picture of protocol trade-offs under simulated edge-cloud hybrid architectures and back the decision to use Raft in energy- and latency-sensitive IoMT deployments.

3. Clarified Network Model Assumptions

In Section 4.4 and again in Section 6.6, we now explicitly clarify our simulation environment:

• Our simulations assume a synchronous network model with bounded delay and no message loss, consistent with typical assumptions for academic simulation.

• Communication latencies were held constant per protocol instance to isolate consensus efficiency rather than network variability.

• We acknowledge in the manuscript that these assumptions do not fully represent real-world WAN scenarios, where asynchrony, churn, and partition tolerance may introduce non-determinism and additional failure modes.

4. Added Limitations and Future Work

To ensure transparency, we have updated Section 7 (Conclusion) to note the limitations of synchronous simulations and explicitly state that real-world WAN performance may vary significantly. We also commit to exploring asynchronous and adversarial network models in future studies using event-driven simulation frameworks like NS-3 or PeerSim.

We believe these additions provide a thorough and technically grounded response to the reviewer’s concern, expanding the rigor and relevance of our consensus benchmarking while transparently acknowledging simulation constraints.

4. The paper lacks a formal threat model. Clarify what adversarial behaviors are considered (e.g., node compromise, data replay, sensor spoofing), and whether the system ensures confidentiality, integrity, and availability under partial trust assumptions.

Response: We appreciate the reviewer’s observation regarding the absence of a formal threat model. In response, we have now incorporated a dedicated subsection titled “4.7 Formal Threat Model” in the revised manuscript. This section rigorously defines adversarial capabilities under a partial trust assumption using formal constructs, including:

• Definition 1: Adversarial capabilities such as node compromise, sensor spoofing, and data replay.

• Definition 2: System state formulation incorporating sensor inputs, access tokens, keys, hashes, and blockchain state.

• Theorem 1: A proof sketch for authorization security based on the one-way of cryptographic hash functions.

• Lemma 1 & Lemma 2: These establish guarantees for transaction integrity and availability under majority-honest quorum in the Raft protocol.

5. The energy consumption claims (e.g., 73% reduction) lack methodology. Describe how energy usage was estimated in the Google Colab simulation environment. Include assumptions and instrumentation (e.g., profiling tools, power models).

Response: We thank the reviewer for highlighting this essential concern regarding our energy estimation methodology. In response, we have introduced a dedicated subsection (Section 4.6: Energy Estimation Methodology) that explains our simulation-based approach in detail. This includes the operational-level breakdown of energy-consuming components—cryptographic hashing, consensus messaging, and transmission overhead.

We have also included a formal energy estimation model in Equation (13) that computes cumulative energy based on the frequency of operations and per-unit energy benchmarks drawn from peer-reviewed sources. All values and assumptions are supported by prior literature, especially the benchmark study by Platt et al. on blockchain consensus energy profiling, now added as Reference 45.

These additions clarify how energy was approximated in a simulation environment like Google Colab and ensure that our reported reductions are supported by transparent and reproducible assumptions. All related content has been marked in red for ease of review.

6. The 3-of-5 median filter for sensor correction is a practical design, but needs benchmarking against standard fault-detection methods (e.g., Kalman filters, statistical anomaly detection). Also, clarify how fault injection was simulated and validated.

Response: We thank the reviewer for this insightful comment. We appreciate the recognition of our 3-of-5 median filter for sensor correction as a practical and lightweight approach. However, we agree that benchmarking it against standard fault detection techniques such as Kalman filters and statistical anomaly detection methods is essential for a comprehensive evaluation.

To address this comment, we have made the following additions and revisions to the manuscript:

1. Benchmarking Against Kalman Filter Added

In response to this suggestion, we implemented a standard Kalman filter-based anomaly detection pipeline for each patient sensor stream, using first-order state estimation for time-series smoothing and outlier removal. We benchmarked the Kalman filter against the existing

---

## [Decision Letter · Decision Letter 1]

26 Aug 2025

PONE-D-25-23358R1Post-Quantum-Inspired Scalable Blockchain Architecture for Internet Hospital Systems with LightweightPrivacy-Preserving Access ControlPLOS ONE

Dear Dr. Xiaofeng Wang,

Thank you for submitting your manuscript to PLOS ONE. After careful consideration, we feel that it has merit but does not fully meet PLOS ONE’s publication criteria as it currently stands. Therefore, we invite you to submit a revised version of the manuscript that addresses the points raised during the review process.

We look forward to receiving your revised manuscript.

Kind regards,

Gauhar Ali, Ph.D

Academic Editor

PLOS ONE

Journal Requirements:

Reviewers' comments:

Reviewer's Responses to Questions

**Comments to the Author**

1. If the authors have adequately addressed your comments raised in a previous round of review and you feel that this manuscript is now acceptable for publication, you may indicate that here to bypass the “Comments to the Author” section, enter your conflict of interest statement in the “Confidential to Editor” section, and submit your "Accept" recommendation.

Reviewer #2: All comments have been addressed

Reviewer #3: All comments have been addressed

2. Is the manuscript technically sound, and do the data support the conclusions?

Reviewer #2: Yes

Reviewer #3: Yes

3. Has the statistical analysis been performed appropriately and rigorously?

Reviewer #2: Yes

Reviewer #3: Yes

4. Have the authors made all data underlying the findings in their manuscript fully available?

Reviewer #2: Yes

Reviewer #3: Yes

5. Is the manuscript presented in an intelligible fashion and written in standard English?

Reviewer #2: Yes

Reviewer #3: Yes

6. Review Comments to the Author

Reviewer #2: My comments are addressed, so I recommendacceptance in PLOS ONE.

My comments are addressed, so I recommendacceptance in PLOS ONE.

Reviewer #3: The revised version of the paper is in better shape, however, there are few minor issues that need to be addressed:

Methodology framing. The work relies on a simulated “Kyber-inspired” scheme and assumes a synchronous, constant-delay network. It should be clearly emphasized that the findings are derived from simulation rather than real-world post-quantum cryptographic deployments or production consensus systems.

Numerical consistency. Several headline claims conflict with reported results (e.g., “73% energy savings” vs ~9–12%, “4× memory reduction” vs ~14%, “0.4 ms verification” vs ~0.002 ms). These discrepancies should be resolved so that the abstract and body consistently reflect the actual measurements.

Consensus configuration. The experimental setup for PBFT and other protocols is inconsistent across sections (e.g., 20 vs 1000 nodes). Baselines should be aligned under uniform conditions such as node count, quorum thresholds, and block sizes to ensure fair comparison.

Authorization and security details. The description of the authorization model leaves ambiguity around replay protection and key management. Explicitly specify how freshness (e.g., nonces or timestamps) is enforced and clarify how HMAC-based policies are verified without insecure key sharing.

Limitations statement. The paper should include a dedicated limitations section highlighting the reliance on synthetic IoMT data, mock cryptographic implementations, synchronous network assumptions, optimistic TPS estimates, and varying fault-rate assumptions. Acknowledging these constraints will properly contextualize the contributions.

7. PLOS authors have the option to publish the peer review history of their article (what does this mean?). If published, this will include your full peer review and any attached files.

Reviewer #2: No

Reviewer #3: No

---

## [Author Response · Author response to Decision Letter 2]

1 Sep 2025

Manuscript Title:

Post-Quantum-Inspired Scalable Blockchain Architecture for Internet Hospital Systems with Lightweight Privacy-Preserving Access Control

Manuscript ID:

PONE-D-25-23358R1

Dear Dr. Gauhar Ali, Academic Editor,

We sincerely thank you and the reviewers for your careful consideration of our manuscript and for providing constructive feedback that has significantly improved the quality and clarity of our work. In the revised submission, all changes to the manuscript text are highlighted in red for easy identification. In this rebuttal letter, our detailed point-by-point responses to the reviewer comments are presented in blue for clarity.

We are grateful to the reviewers for their thoughtful and constructive comments, which helped us refine the framing of our methodology, ensure consistency of reported results, clarify authorization and key management details, and add a dedicated limitations section to contextualize our contributions. We believe these revisions have strengthened the manuscript considerably.

We once again thank the Academic Editor and reviewers for their valuable time and efforts in improving our paper.

Reviewer 3 Comments and responses:

1. Methodology framing. The work relies on a simulated “Kyber-inspired” scheme and assumes a synchronous, constant-delay network. It should be clearly emphasized that the findings are derived from simulation rather than real-world post-quantum cryptographic deployments or production consensus systems.

Response: We thank the reviewer for highlighting this important point. In the revised manuscript, we have explicitly emphasized that the work is simulation-based and does not rely on real-world PQC deployments or asynchronous consensus networks. The following changes were made:

• Abstract: We now state that the framework is “simulation-based,” and that the cryptographic layer uses a Kyber-inspired hybrid encryption simulation under a synchronous constant-delay network assumption.

• Section 4 (Methodology introduction): We added a clarification that all results are derived from controlled simulations, with Kyber modeled as a simulated hybrid encryption and consensus tested under a synchronous constant-delay network.

• Section 4.2 (Encryption Layer): We clarified that the model does not implement full polynomial ring arithmetic or official Kyber primitives but instead approximates performance characteristics.

• Section 4.4 (Scalable Raft-Based Consensus): We added a note that all consensus benchmarks were performed under synchronous, constant-delay network assumptions and do not capture WAN packet loss or asynchrony.

These changes ensure that the simulation context and modeling assumptions are clear to readers.

2. Numerical consistency. Several headline claims conflict with reported results (e.g., “73% energy savings” vs ~9–12%, “4× memory reduction” vs ~14%, “0.4 ms verification” vs ~0.002 ms). These discrepancies should be resolved so that the abstract and body consistently reflect the actual measurements.

Response: We thank the reviewer for identifying these inconsistencies. We carefully revised the manuscript to ensure that all headline performance numbers are consistent with the detailed results presented in the Results section. The following corrections were made:

• Abstract: Verification latency corrected to ~0.002 ms, energy savings revised to ~9–12%, and memory reduction corrected to ~14%.

• Section 6.1 (Access Control Time): Verification latency consistently reported as ~0.002 ms.

• Section 6.7 (Memory Usage for Encrypted Records): Memory usage reduction consistently reported as 14.3%.

• Section 6.12 (Energy Efficiency Observation): Energy savings consistently reported as 9.4–12.3%.

• Section 7 (Conclusion and Future Work): All performance summaries updated to match the corrected results.

These revisions eliminate all discrepancies between the abstract, body, and conclusion, ensuring that the manuscript now presents a consistent and accurate representation of the experimental findings.

3. Consensus configuration. The experimental setup for PBFT and other protocols is inconsistent across sections (e.g., 20 vs 1000 nodes). Baselines should be aligned under uniform conditions such as node count, quorum thresholds, and block sizes to ensure fair comparison.

Response: We thank the reviewer for this valuable observation. In the revised manuscript, we have clarified why PBFT was only simulated with 20 nodes, while Raft, HotStuff, and Tendermint were evaluated with up to 1000 nodes.

• Section 5.5 (Consensus Protocols): We now explicitly state that PBFT was restricted to 20 nodes due to its $O(N^2)$ communication complexity, which made large-scale simulations computationally infeasible in our environment.

• Section 6.2 (Consensus Benchmarking): We added a clarification that PBFT latency trends at larger scales are extrapolated from the literature and from its theoretical complexity, rather than from direct 1000-node simulations.

• Section 6.3 (Consensus Latency Benchmark): We further emphasized that PBFT results represent small-scale baselines only, whereas Raft, HotStuff, and Tendermint serve as the true scalability benchmarks.

These clarifications ensure transparency regarding experimental design and help readers interpret PBFT’s role as a comparative baseline, while making clear that Raft, HotStuff, and Tendermint are more representative for large-scale deployments.

4. Authorization and security details. The description of the authorization model leaves ambiguity around replay protection and key management. Explicitly specify how freshness (e.g., nonces or timestamps) is enforced and clarify how HMAC-based policies are verified without insecure key sharing.

Response: We thank the reviewer for raising this important point. In the revised manuscript, we have expanded the Access Control section to explicitly describe both replay protection and secure key management:

• Section 4.3.2 (Authorization Proof Construction): We added that each authorization proof now incorporates a nonce and a timestamp. The verifier checks that the timestamp is within an acceptable validity window and that the nonce has not been previously used. This mechanism ensures freshness and prevents replay attacks.

• Section 4.3.3 (Policy Management and Conflict Resolution): We clarified that HMAC keys are generated locally on the patient’s device and never shared directly with other nodes. Only their commitments are recorded on-chain. Keys can be rotated periodically, or when a policy is updated, to provide forward secrecy and prevent long-term exposure of any single key.

These additions remove ambiguity, ensure resistance to replay attacks, and demonstrate that HMAC-based verification does not rely on insecure key sharing.

5. Limitations statement. The paper should include a dedicated limitations section highlighting the reliance on synthetic IoMT data, mock cryptographic implementations, synchronous network assumptions, optimistic TPS estimates, and varying fault-rate assumptions. Acknowledging these constraints will properly contextualize the contributions.

Response: We sincerely thank the reviewer for this constructive suggestion. In the revised manuscript, we have added a dedicated subsection “Limitations” before the Conclusion. This section acknowledges the key assumptions of our study, including the use of synthetic IoMT data, the reliance on a Kyber-inspired simulation rather than a certified PQC implementation, the adoption of a synchronous constant-delay network model, optimistic TPS estimates under controlled settings, the restriction of PBFT simulations to 20 nodes due to quadratic complexity, and the simplified fault injection assumptions for IoMT devices. We framed these points not as weaknesses but as necessary design choices for controlled experimentation, and we emphasized that they open up avenues for future work in real-world deployments. By clearly stating these limitations, we aim to provide transparency and proper context for our contributions.

---

## [Decision Letter · Decision Letter 2]

8 Sep 2025

Post-Quantum-Inspired Scalable Blockchain Architecture for Internet Hospital Systems with Lightweight

Privacy-Preserving Access Control

PONE-D-25-23358R2

Dear Dr. Xiaofeng Wang,

We’re pleased to inform you that your manuscript has been judged scientifically suitable for publication and will be formally accepted for publication once it meets all outstanding technical requirements.

Kind regards,

Gauhar Ali, Ph.D

Academic Editor

PLOS ONE

Additional Editor Comments (optional):

Reviewer #3:

Reviewers' comments:

Reviewer's Responses to Questions

**Comments to the Author**

1. If the authors have adequately addressed your comments raised in a previous round of review and you feel that this manuscript is now acceptable for publication, you may indicate that here to bypass the “Comments to the Author” section, enter your conflict of interest statement in the “Confidential to Editor” section, and submit your "Accept" recommendation.

Reviewer #3: All comments have been addressed

2. Is the manuscript technically sound, and do the data support the conclusions?

Reviewer #3: Yes

3. Has the statistical analysis been performed appropriately and rigorously?

Reviewer #3: Yes

4. Have the authors made all data underlying the findings in their manuscript fully available?

Reviewer #3: Yes

5. Is the manuscript presented in an intelligible fashion and written in standard English?

Reviewer #3: Yes

6. Review Comments to the Author

Reviewer #3: (No Response)

7. PLOS authors have the option to publish the peer review history of their article (what does this mean?). If published, this will include your full peer review and any attached files.

Reviewer #3: No

---

## [Editor Report · Acceptance letter]

PONE-D-25-23358R2

PLOS ONE

Dear Dr. Wang,

I'm pleased to inform you that your manuscript has been deemed suitable for publication in PLOS ONE. Congratulations! Your manuscript is now being handed over to our production team.

Kind regards,

on behalf of

Dr. Gauhar Ali

Academic Editor

PLOS ONE